# Explicit loss asymptotics in the gradient descent training of neural networks

**Maksim Velikanov**
Skolkovo Institute of Science and Technology
`maksim.velikanov@skoltech.ru`

**Dmitry Yarotsky**
Skolkovo Institute of Science and Technology
`d.yarotsky@skoltech.ru`

## Abstract

Current theoretical results on optimization trajectories of neural networks trained by gradient descent typically have the form of rigorous but potentially loose bounds on the loss values. In the present work we take a different approach and show that the learning trajectory of a wide network in a lazy training regime can be characterized by an explicit asymptotic at large training times. Specifically, the leading term in the asymptotic expansion of the loss behaves as a power law $L(t) \sim Ct^{-\xi}$ with exponent $\xi$ expressed only through the data dimension, the smoothness of the activation function, and the class of function being approximated. Our results are based on spectral analysis of the integral operator representing the linearized evolution of a large network trained on the expected loss. Importantly, the techniques we employ do not require a specific form of the data distribution, for example Gaussian, thus making our findings sufficiently universal.

## 1   Introduction

A major challenge in the research of neural networks is the quantitative theoretical description of their optimization by gradient descent. At present, many aspects of network training seem to be understood rather well on a qualitative level, or admit convincing heuristic explanations, but we seem to lack tools for making reasonably accurate quantitative predictions, even for relatively simple models and data. In this sense, the theory of neural networks compares unfavorably to physics, which is also an application-driven field but with an apparently much more successful penetration of theoretical methods. The main difficulty here is probably the complex structure of the data and models, which are hard to describe in terms of convenient and simple mathematical abstractions.

In recent years, a significant progress in the theoretical analysis of gradient descent of neural networks has been associated with the limit of large networks, which can be studied using various methods from partial differential equations [26, 31], kernel methods [20, 21], spin glass theory [12], random matrix theory [29], dynamical systems [30], and other mathematical fields.

In the present work, we consider a setting of large networks and large, smoothly distributed data sets that allows us to obtain explicit leading terms in the long-term evolution of the loss under gradient descent. We are inspired by the spectral theory of singular integral operators [5], which we apply to the linearized evolution of the network. While this linearized evolution has been widely studied recently, most related research seems to focus on theoretical convergence guarantees and upper bounds for the loss values [3, 27], or on a highly symmetric problems admitting explicit solution [35, 38]. In contrast, we focus on explicit loss evolution formulas, which we find as power laws

$$L(t) \sim Ct^{-\xi}. \tag{1}$$

We argue that the exponents $\xi$ here exhibit some form of universality, in that they are essentially determined by the input dimension $d$ and by the smoothness classes of the activation function and the target function. In particular, we find that in the case of ReLU networks approximating an indicator

35th Conference on Neural Information Processing Systems (NeurIPS 2021).

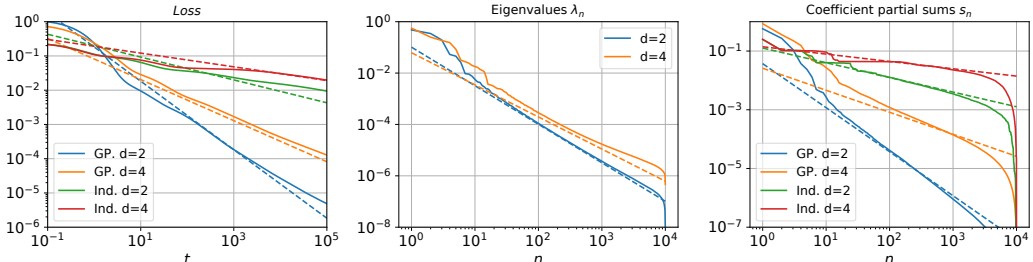

Figure 1: The loss trajectories and spectral properties of the neural tangent kernels of shallow networks in the NTK regime. The target function (i.e., the initial displacement between the network output and the approximated function) is either generated by a Gaussian process (*GP*) modeled by a larger network of the same architecture, or is an indicator function of a $d$-dimensional ball (*Ind*). The data distributions $\mu$ are modeled as mixtures of 8 Gaussian distributions with random centers, and the data dimension is either $d = 2$ or $d = 4$. The **solid** lines show the numerically obtained values, while the **dashed** lines show the respective theoretical power-law asymptotics. The dataset size is $M = 10^4$ (see Section A (SM) for further details of experiments).
**Left:** Loss evolution for a shallow network with width $N = 3000$. The scaling exponent giving the slope of the theoretical asymptotic is $\xi = \frac{\beta}{d+\alpha} = \frac{3}{d+1}$ for GP and $\xi = \frac{1}{d+\alpha}$ for Ind (see Section 5.2).
**Center:** Distribution of the infinite network NTK eigenvalues $\lambda_n$. The theoretical scaling exponent is $\nu = 1 + \frac{1}{d}$ (see Section 5.1). **Right:** Distributions of the coefficient partial sums $s_n$ (see Eq. (7)). The theoretical scaling exponent is $\kappa = \frac{\beta}{d} = \frac{3}{d}$ for GP and $\kappa = \frac{1}{d}$ for Ind (see Section 5.2).

function of some region in the $d$-dimensional space (a classification problem target), the natural value of the exponent is $\xi = \frac{1}{d+1}$. On the other hand, in the case of target functions generated by a randomly initialized wide ReLU network, the exponent is $\xi = \frac{3}{d+1}$. Our approach also allows us to obtain explicit expressions for the coefficient $C$ in these cases.

The power law (1) is established using similar power laws (but with different exponents and coefficients) for the eigenvalues of the evolution operator and for the coefficients in the expansion of the target function over corresponding eigenvectors. These power laws are indeed confirmed by our experiments (see Figure 1).

Our main scenario is approximation by shallow ReLU network in the NTK regime, but we also briefly consider several modifications of this scenario, namely the activation functions $(x_+)^q$ with $q > 0$, approximation by a deep network in the NTK regime, and approximation in the mean field regime.

## 2 Related work

The approximation of linearized network evolution and its applications were studied in many works, see in particular [21, 10, 23, 22, 19]. The role of the eigenvalues and eigenvectors of the NTK in the linearized network was emphasized in [3], where the GD dynamic of the finite network was linked to the dynamic of its infinite width counterpart, determined by spectral properties of the corresponding NTK. In subsequent works, the NTK spectrum was central for description of network training [27] and generalization [7, 4]. These papers use the assumption of power law NTK spectrum, but justify it empirically or for highly symmetric problems.

Because of the importance of the NTK spectrum, a number of works focused on its description in different settings. We first mention directions and settings that are different from ours. The case of very deep networks was studied in [37, 18, 17, 2]. This analysis is relying on convergence of the NTK to simple fixed points in the limit of infinite depth. [17] also studied the choice of activation function, in particular its smoothness. Another line of research [1, 13] uses techniques from Random Matrix Theory to analyze the setting where the dataset size $M$ goes to infinity together with data dimension $d$ and layer widths $n_l$.

In this work we consider the setting of fixed data dimension and effectively infinite network width and dataset size. In this case [35] showed that the network evolution can be described by a deterministic

integral operator that is easier to analyze than a large but finite matrix. Also, in that paper and in the papers [38, 8, 27], the integral operator was explicitly diagonalized in the special case of uniform distribution on a sphere.

A power law upper bound for the training loss was obtained in [27], but with an exponent $\xi$ smaller than ours. The paper [4] describes a different power law, relating the test loss at the end of training to the dataset size and the network width. This result also relies on a power law assumption for the NTK spectrum.

Another related line of research is the case of univariate functions. The gradient descent evolution of univariate shallow networks has been analytically studied in the papers [39, 36].

## 3 Asymptotic evolution of the loss function

We consider a linearized training of a neural network by gradient descent. Such linear approximations arise naturally in various "lazy training" scenarios [10]. Consider the standard quadratic loss function $L(\mathbf{W}) = \frac{1}{2} \int_{\mathbb{R}^d} |\widetilde{f}(\mathbf{W}, \mathbf{x}) - f(\mathbf{x})|^2 \mu(\mathbf{x}) d\mathbf{x}$, where $\mathbf{x}$ is the $d$-dimensional input, $f$ is the approximated function, $\widetilde{f}$ is the network, $\mathbf{W}$ are the network weights, and $\mu$ is the data distribution on which the network is trained. Gradient descent can be written as the differential equation $\frac{d}{dt} \mathbf{W} = -\nabla_{\mathbf{W}} L(\mathbf{W})$. We assume that the weight vector $\mathbf{W}$ is close to a global minimum $\mathbf{W}_*$ where $L(\mathbf{W}_*) = 0$ and $\widetilde{f}(\mathbf{W}_*, \mathbf{x}) = f(\mathbf{x})$ $\mu$-a.e., and that the evolution equation can be linearized at $\mathbf{W} = \mathbf{W}_*$. It is convenient to write this linearized equation in terms of the difference $\delta f(\mathbf{x}) = \widetilde{f}(\mathbf{W}, \mathbf{x}) - f(\mathbf{x})$ between the current output and the target. The corresponding linear equation is $\frac{d}{dt} \delta f = -\mathcal{A} \delta f$, where $\mathcal{A}$ can be written as the integral operator $\mathcal{A} \delta f(\mathbf{x}) = \int_{\mathbb{R}^d} \Theta(\mathbf{x}, \mathbf{x}') \mu(\mathbf{x}') \delta f(\mathbf{x}') d\mathbf{x}'$ with the NTK (neural tangent kernel)

$$\Theta(\mathbf{x}, \mathbf{x}') = \nabla_{\mathbf{W}} \widetilde{f}(\mathbf{W}_*, \mathbf{x})^T \nabla_{\mathbf{W}} \widetilde{f}(\mathbf{W}_*, \mathbf{x}') \tag{2}$$

The evolution operator $\mathcal{A}$ is a symmetric non-negative definite operator with respect to the scalar product $\langle f_1, f_2 \rangle_\mu = \int f_1(\mathbf{x}) f_2(\mathbf{x}) \mu(\mathbf{x}) d\mathbf{x}$. By multiplying functions $f$ by $\mu^{1/2}$, the operator $\mathcal{A}$ can be brought to the form $\widetilde{\mathcal{A}} = \mu^{1/2} \mathcal{A} \mu^{-1/2}$ with a symmetric kernel,

$$\widetilde{\mathcal{A}} \delta f(\mathbf{x}) = \int_{\mathbb{R}^d} \mu^{1/2}(\mathbf{x}) \Theta(\mathbf{x}, \mathbf{x}') \mu^{1/2}(\mathbf{x}') \delta f(\mathbf{x}') d\mathbf{x}'. \tag{3}$$

In this form $\widetilde{\mathcal{A}}$ is symmetric w.r.t. the usual scalar product $\langle f_1, f_2 \rangle = \int f_1(\mathbf{x}) f_2(\mathbf{x}) d\mathbf{x}$. Observe that the loss at time $t$ can be written as $L(t) = \frac{1}{2} \|g_t\|^2$, where $g_t = e^{-t\widetilde{\mathcal{A}}} g$, the norm $\|\cdot\|$ corresponds to the scalar product $\langle \cdot, \cdot \rangle$, and the function $g$ is given by

$$g(\mathbf{x}) = \mu^{1/2}(\mathbf{x})(\widetilde{f}(\mathbf{w}(t=0), \mathbf{x}) - f(\mathbf{x})). \tag{4}$$

We can now describe the evolution of the loss by diagonalizing the operator $\widetilde{\mathcal{A}}$. We will be interested in the scenario where $\mu$ is a smooth function that is compactly supported or falls off at infinity sufficiently fast. (In particular, in the context of a finite training set this means that this set is large enough to be legitimately approximated by $\mu$.) In this case, for typical kernels $\Theta$, the operator $\widetilde{\mathcal{A}}$ will have a discrete spectrum with eigenvalues converging to 0. Let $\lambda_n$ denote the eigenvalues of $\widetilde{\mathcal{A}}$ sorted in decreasing order, and let $c_n$ be the respective coefficients in the expansion of the initial error $g$ (given by Eq. (4)) over the normalized orthogonal eigenvectors. Then, the loss evolves by

$$L(t) = \frac{1}{2} \sum_{n=0}^{\infty} e^{-2\lambda_n t} |c_n|^2. \tag{5}$$

To compute the asymptotic of $L(t)$ at large times $t$, we need to know the distribution of the eigenvalues $\lambda_n$ and the coefficients $c_n$ at large $n$. The key assumption of our work (verified later for certain scenarios) is that these distributions have power law forms. Specifically, regarding the eigenvalues $\lambda_n$ we assume that

$$\lambda_n \sim \Lambda n^{-\nu} \tag{6}$$

with some coefficient $\Lambda$ and exponent $\nu$. Regarding the coefficients $c_n$, we assume that they also have a power law distribution on a large length scale in $n$, but possibly deviate from this law locally (e.g., due to oscillations). For this reason, we describe their large $n$ behavior by the partial sums

$$s_n \equiv \sum_{k \geq n} |c_k|^2. \tag{7}$$

We then assume that, for some coefficient $K$ and exponent $\kappa$,

$$s_n \sim K n^{-\kappa}. \tag{8}$$

Under assumption of the power laws (6) and (8), it is easy to check (see SM, Section B) that the loss also has a power law asymptotic (1) with the constant $C$ and exponent $\xi$ expressible through the constants $\Lambda, K$ and exponents $\nu, \kappa$ :

$$L(t) \sim \tfrac{K}{2} \Gamma \left( \tfrac{\kappa}{\nu} + 1 \right) (2\Lambda t)^{-\frac{\kappa}{\nu}}, \tag{9}$$

where $\Gamma(z)$ is the Gamma function.

In Figure 1 we illustrate this approach to the long-term loss evolution with several examples of target functions having different smoothness and dimension and, as a result, exhibiting different exponents.

In the remainder of the paper we show that the power laws (6) and (8), and hence the large-$t$ asymptotic (9) of the loss, are indeed valid for some natural network training scenarios. The asymptotic (6) of the eigenvalues is primarily determined by the singularities of the kernel $\Theta$. These singularities can be explicitly described for shallow neural network with piecewise smooth activations such as ReLU. Then, the power law (6) can be derived from general results on integral operators with singular kernels. In particular, in the case of ReLU we find that $\nu = 1 + \frac{1}{d}$.

The asymptotic (8) of the coefficients is more subtle, as it depends significantly on the class of the initial error function $g$ (which in turn depends, by Eq. (4), on the target function $f$ and the initial approximation $\widetilde{f}$). We derive this power law for one natural class of discontinuous functions $g$, and for functions $g$ that are realizations of a Gaussian process of a particular "roughness".

Moreover, for these two classes, we also find an explicit form of the coefficient $C$ appearing in loss asymptotic (1). This requires us, however, to modify the above derivation of the loss asymptotic (9) by what can be called "integrated localization". Roughly speaking, in the large-$t$ limit we can think of the eigenvectors of $\widetilde{\mathcal{A}}$ as infinitesimally localized in $\mathbb{R}^d$. We then apply the above derivation of Eq. (9) not to the full set of eigenvectors, but separately to each infinitesimal sub-domain, and then integrate the results. See details in Sections 5.2 and D (SM).

The subsequent exposition is structured as follows. In Section 4 we provide the background on the NTK kernel and spectral properties of singular integral operators. Then, in Section 5, we derive the loss asymptotic (1) for our main setting – the NTK training with a shallow ReLU network. After that, in Section 6 we consider various modifications of this setting: other activation functions (Section 6.1), deep networks (Section 6.2), and training in the mean field regime (Section 6.3).

## 4  Background

### 4.1  Infinitely wide networks

Lazy training scenarios discussed in section 3 naturally arise for the networks in the limit of infinite width. However, there are several ways to scale parameters of the network with width, which lead to different operating regimes of infinitely wide networks [15, 16].

**NTK regime.** The first option we consider is the NTK regime [20], for which the NTK $\Theta$ defined in (2) is deterministic and constant during training. This simplification immediately leads to a linear dynamic in the space of network outputs. We consider feed-forward networks parametrized as

$$\begin{cases} z_j^1 = \sum_{i=1}^{d} \sigma_w w_{ij}^1 x_i + \sigma_b b_j^1 \\ z_j^l = \sum_{i=1}^{n_{l-1}} \frac{\sigma_w}{\sqrt{n_{l-1}}} w_{ij}^l x_i^{l-1} + \sigma_b b_j^l, \quad l > 1 \\ x_j^l = \phi(z_j^l) \end{cases} \tag{10}$$

Here $n^l$ is the width of layer $l$, $x_i$ is a network input and $z_j^L$ is the network output. We also consider the last layer without bias term $b_j^L$ and having width $n_L = 1$ (scalar output). Trainable parameters $w_{ij}^l$, $b_j^l$ are initialized as i.i.d. normal Gaussians.

The output of each layer $l$ at initialisation is a Gaussian process with covariance $\langle z_j^l(\mathbf{x}) z_{j'}^l(\mathbf{x}') \rangle = \delta_{jj'} \Sigma^{(l)}(\mathbf{x}, \mathbf{x}')$. By introducing the NTK's $\Theta^{(l)}(\mathbf{x}, \mathbf{x}')$ of intermediate layers, one can recursively compute [21] the both NTK and covariance

$$
\begin{cases}
\Sigma^{(l+1)} = \sigma_w^2 \langle \phi(z^l) \phi(z'^l) \rangle + \sigma_b^2 \\
\Theta^{(l+1)} = \Sigma^{(l+1)} + \sigma_w^2 \Theta^{(l)} \langle \dot{\phi}(z^l) \dot{\phi}(z'^l) \rangle
\end{cases}
\tag{11}
$$

Here $z^l = z^l(\mathbf{x})$ and $z'^l = z^l(\mathbf{x}')$ are draws from the Gaussian process with covariance $\Sigma^{(l)}(\mathbf{x}, \mathbf{x}')$, and $\langle \ldots \rangle$ is the averaging. It is also convenient to parametrize the covariance at input points $\mathbf{x}, \mathbf{x}'$ in some layer $l$ as

$$
\Sigma^{(l)}(\mathbf{x}, \mathbf{x}') = \begin{pmatrix} r_l^2 & r_l r_l' \cos \varphi_l \\ r_l r_l' \cos \varphi_l & r_l'^2 \end{pmatrix}
\tag{12}
$$

To analyze the spectrum of the evolution operator $\widetilde{\mathcal{A}}$ given by (3) we will intensively use the explicit form of the NTK and covariance. For the ReLU activation $\phi(z) = (z)_+$ the averages in (11) can be computed analytically [11]. In the case of shallow network ($L = 2$) the result is:

$$
\Sigma(\mathbf{x}, \mathbf{x}') = \frac{\sigma_w^2}{2\pi} r r' \big( \sin \varphi + \cos \varphi (\pi - \varphi) \big)
\tag{13}
$$

$$
\Theta(\mathbf{x}, \mathbf{x}') = \Sigma(\mathbf{x}, \mathbf{x}') + \frac{\sigma_w^2}{2\pi} r r' \cos \varphi (\pi - \varphi)
\tag{14}
$$

Here $r, r', \varphi$ are parameters from (12) with $l = 1$ (we dropped index 1). They have a clear interpretation in terms of extended input vectors $\tilde{\mathbf{x}} = (\sigma_w \mathbf{x}, \sigma_b) \in \mathbb{R}^{d+1}$. Specifically, $r = \|\tilde{\mathbf{x}}\|$, $r' = \|\tilde{\mathbf{x}}'\|$ and $\varphi$ is the angle between $\tilde{\mathbf{x}}$ and $\tilde{\mathbf{x}}'$.

**Mean Field regime.** This operating regime of infinitely wide networks is naturally defined [26, 31] for shallow networks of the form $f(\mathbf{W}, \mathbf{x}) = \frac{1}{N} \sum_{i=1}^N c_i \phi(\mathbf{w}_i \cdot \mathbf{x} + b_i) = \frac{1}{N} \sum_{i=1}^N \tilde{\phi}(\tilde{\mathbf{w}}_i, \mathbf{x})$. Here $\tilde{\mathbf{w}}_i = (c_i, \mathbf{w}_i, b_i)$ denotes the collection of parameters, associated with a single neuron. In the infinite width limit $N \to \infty$ the evolution is described by a PDE on parameter density distribution $p(\tilde{\mathbf{w}})$:

$$
\partial_t p = \nabla_{\tilde{\mathbf{w}}} \Big[ p \nabla_{\tilde{\mathbf{w}}} \int K(\tilde{\mathbf{w}}, \tilde{\mathbf{w}}')(p(\tilde{\mathbf{w}}') - p_\infty(\tilde{\mathbf{w}}')) d\tilde{\mathbf{w}}' \Big], \quad K(\tilde{\mathbf{w}}, \tilde{\mathbf{w}}') = \int \tilde{\phi}(\tilde{\mathbf{w}}, \mathbf{x}) \mu(\mathbf{x}) \tilde{\phi}(\tilde{\mathbf{w}}', \mathbf{x}) d\mathbf{x}
$$

Under mild assumptions [9], solution of the MF equation converges to the global optimum $p_\infty(\tilde{\mathbf{w}}')$. At large times $t$ the MF equation can be linearized [39] around $p_\infty$, thus bringing the network to the lazy training regime discussed in section 3. The network NTK in this case is equal to

$$
\Theta(\mathbf{x}, \mathbf{x}') = \int \nabla_{\tilde{\mathbf{w}}} \tilde{\phi}(\tilde{\mathbf{w}}, \mathbf{x}) p_\infty(\tilde{\mathbf{w}}) \nabla_{\tilde{\mathbf{w}}} \tilde{\phi}(\tilde{\mathbf{w}}, \mathbf{x}') d\tilde{\mathbf{w}}.
\tag{15}
$$

### 4.2 Singular integral operators

Consider the evolution operator $\widetilde{\mathcal{A}}$ given by Eq. (3). Under our assumptions, the kernel $\mu^{1/2}(\mathbf{x}) \Theta(\mathbf{x}, \mathbf{x}') \mu^{1/2}(\mathbf{x}')$ of this operator quickly falls off at infinity and is smooth outside the diagonal $\mathbf{x} = \mathbf{x}'$, but, as we will see later, has a homogeneous singularity on this diagonal. In this setting, a general theory developed in [5] allows us to obtain the asymptotic distribution (6) of the eigenvalues with explicit constants $\Lambda$ and $\nu$.

Specifically, suppose that in a neighborhood of the diagonal the kernel $\Theta$ has a representation

$$
\Theta(\mathbf{x}, \mathbf{x}') = \theta_{\mathbf{x}}(\mathbf{x} - \mathbf{x}') + \ldots,
\tag{16}
$$

where $\theta_{\mathbf{x}}(\cdot)$ is a (possibly $\mathbf{x}$-dependent) even ($\theta_{\mathbf{x}}(\mathbf{z}) = \theta_{\mathbf{x}}(-\mathbf{z})$) homogeneous function of degree $\alpha$:

$$
\theta_{\mathbf{x}}(c\mathbf{z}) = |c|^\alpha \theta_{\mathbf{x}}(\mathbf{z}),
\tag{17}
$$

and the dots $\ldots$ denote terms of higher smoothness.

Let $N_\lambda$ denote the number of eigenvalues of $\widetilde{\mathcal{A}}$ greater than $\lambda$. Then, it is shown in [5] that for small $\lambda$, the leading term of $N_\lambda$ is given by

$$N_\lambda \sim \Big( \int \gamma_{\mathbf{x}} \mu^{\frac{d}{d+\alpha}}(\mathbf{x}) d\mathbf{x} \Big) \lambda^{-\frac{d}{d+\alpha}}. \tag{18}$$

Here, $\gamma_{\mathbf{x}} = (2\pi)^{-d} |\{\mathbf{k} \in \mathbb{R}^d : \widetilde{\theta}_{\mathbf{x}}(\mathbf{k}) > 1\}|$, where $|\cdot|$ denotes the Lebesgue measure and $\widetilde{\theta}_{\mathbf{x}}$ is a suitably defined Fourier transform of the homogeneous function $\theta_{\mathbf{x}}$ (see SM, Section C).

Formula (18) can be derived as follows. Divide the domain $\mathbb{R}^d$ into multiple small subsets $\Omega_m$, and think of $\widetilde{\mathcal{A}}$ as an operator matrix corresponding to the decomposition $L^2(\mathbb{R}^d) = \oplus_m L^2(\Omega_m)$. Using the fall off and smoothness of the kernel outside the diagonal, one can show that the leading term of $N_\lambda$ is determined only by the diagonal elements of this operator matrix. Then, the leading term can be found by considering each restriction $\widetilde{\mathcal{A}}|_{L^2(\Omega_m)}$ separately and summing the respective contributions to $N_\lambda$, i.e., $N_\lambda(\widetilde{\mathcal{A}}) \sim \sum_m N_\lambda(\widetilde{\mathcal{A}}|_{L^2(\Omega_m)})$. If we decrease the size of each $\Omega_m$ by a factor $M$, then the number of terms in this sum increases $M^d$-fold, but at the same time each $N_\lambda(\widetilde{\mathcal{A}}|_{L^2(\Omega_m)})$ decreases also roughly $M^d$-fold, due to rescaling of eigenvalues. In the limit of infinitely small subsets $\Omega_m$, the operator $\widetilde{\mathcal{A}}|_{L^2(\Omega_m)}$ can be approximated by the convolution with the homogeneous function $\theta_{\mathbf{x}}$. The asymptotic form of its eigenvalues can then be written in terms of the Fourier transform $\widetilde{\theta}_{\mathbf{x}}$ as given above. The power $\frac{d}{d+\alpha}$ in Eq. (18) can be deduced by observing that the volume of the $\mathbf{k}$-space corresponding to $\widetilde{\theta}_{\mathbf{x}} > \lambda$ scales as $\lambda^{-\frac{d}{d+\alpha}}$.

The formula (18) can be translated into the power law (6) by inverting the relation between $\lambda$ and $n$ (note that $N_{\lambda_n} = n$ for any $n$). Specifically, we find that the law (6) holds with

$$\nu = 1 + \frac{\alpha}{d}, \quad \Lambda = \Big( \int \gamma_{\mathbf{x}} \mu^{1/\nu}(\mathbf{x}) d\mathbf{x} \Big)^\nu. \tag{19}$$

In Section 5.2 we will show that this approach can be extended to yield the loss asymptotic (1).

# 5 Asymptotic analysis of wide networks

## 5.1 NTK operators and their singularities

In this section we demonstrate the asymptotic law (6) of the NTK spectrum as well as outputs covariance spectrum, and find constants $\nu, \Lambda$. We consider a shallow ReLU network in the NTK regime and with the data distribution $\mu(\mathbf{x})$ as described in Section 3. The NTK $\Theta(\mathbf{x}, \mathbf{x}')$ and outputs covariance $\Sigma(\mathbf{x}, \mathbf{x}')$ of such network is given by (14), (13) with $r, r'$ and $\varphi$ explicitly depending on input points $\mathbf{x}, \mathbf{x}'$:

$$r(\mathbf{x}) = \sqrt{\sigma_w^2 |\mathbf{x}|^2 + \sigma_b^2}, \quad \varphi(\mathbf{x}, \mathbf{x}') = \arccos\Big(\frac{\sigma_w^2 \mathbf{x} \cdot \mathbf{x}' + \sigma_b^2}{r(\mathbf{x}) r(\mathbf{x}')}\Big). \tag{20}$$

To use the spectral theory described in section 4.2 we analyze the smoothness of the kernels $\Theta$ and $\Sigma$. Firstly, they are smooth (infinitely differentiable) functions of $r, r', \varphi$ on the whole domain. Now suppose that the bias term is not absent: $\sigma_b > 0$. Then $r(\mathbf{x})$ is a smooth function for all $\mathbf{x} \in \mathbb{R}^d$. The argument of $\arccos$ in (20) is also smooth everywhere, but $\arccos(z)$ itself is smooth on $(-1, 1)$ and has divergent derivative at the end points $z = 1, -1$ corresponding to $\varphi = 0, \pi$. We see that condition $\sigma_b > 0$ implies that the case $\varphi = \pi$ is never realized, while $\varphi(\mathbf{x}, \mathbf{x}') = 0$ at all coinciding inputs. Thus we established that $\Theta(\mathbf{x}, \mathbf{x}')$ and $\Sigma(\mathbf{x}, \mathbf{x}')$ are smooth everywhere except the diagonal $\mathbf{x} = \mathbf{x}'$, where they might have a singularity.

To analyze the behavior at the diagonal, we first expand $\varphi$ in Eq. (20) for small $\delta\mathbf{x} = \mathbf{x} - \mathbf{x}'$:

$$\varphi(\mathbf{x}, \mathbf{x}') = \frac{\sigma_w \sqrt{r^2(\mathbf{x}) - \sigma_w^2 |\mathbf{x}|^2 \cos^2 \psi}}{r^2(\mathbf{x})} |\delta\mathbf{x}| + O(|\delta\mathbf{x}|^2) \tag{21}$$

Here $\psi$ is the angle between $\delta\mathbf{x}$ and $\mathbf{x}$. If $\sigma_b > 0$, then the expression under the square root is always positive, therefore the angle $\varphi(\mathbf{x}, \mathbf{x}')$ has a homogeneous singularity of degree 1 on the diagonal. Now we expand expressions (13) and (14) for small $\varphi$ and find that both NTK and covariance indeed have a singularity on the diagonal with leading singular terms.

$$\Sigma_{\text{sing}}(\mathbf{x}, \mathbf{x}') = \frac{\sigma_w^2}{2\pi} r^2(\mathbf{x}) \frac{1}{3} \varphi^3(\mathbf{x}, \mathbf{x}') \propto |\delta\mathbf{x}|^3 \tag{22}$$

$$\Theta_{\text{sing}}(\mathbf{x}, \mathbf{x}') = -\frac{\sigma_w^2}{2\pi} r^2(\mathbf{x}) \varphi(\mathbf{x}, \mathbf{x}') \propto |\delta\mathbf{x}| \tag{23}$$

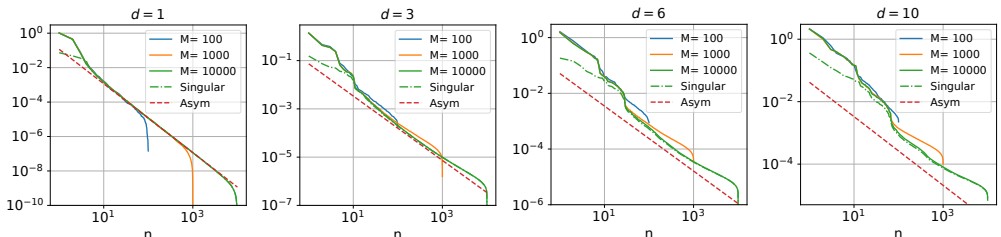

Figure 2: NTK eigenvalues $\lambda_n$ for networks with different input dimension $d$ and for different data set sizes $M$. *Asym* shows the theoretical power law (6) with parameters $\Lambda, \nu$ derived in Section 5.1; in particular, $\nu = 1 + \frac{1}{d}$. *Singular* corresponds to eigenvalues of the singular part of NTK (23). We see that for moderately big $n$ the singular part has the same eigenvalues as the full NTK. Observe that the number $n$ at which the spectrum converges to its asymptotic form increases with dimension $d$.

Now we see that both NTK $\Theta$ and covariance $\Sigma$ have the form needed for application of spectral theory discussed in section 4.2. For NTK we have singularity degree $\alpha = 1$, which by Eq. (19) leads to the already announced exponent $\nu = 1 + \frac{1}{d}$ (see Figure 1). Interestingly, the singularity degree for the covariance is higher (namely, 3), resulting in a faster fall off of the corresponding eigenvalues. In the sequel (see Section 5.2), this latter degree will appear in the analysis of loss asymptotic for target functions generated by the neural network Gaussian process, and will be denoted by $\beta$.

In the case of NTK $\Theta(\mathbf{x}, \mathbf{x}')$ the singularity is essentially given by (21). The corresponding Fourier transform $\widetilde{\theta}_{\mathbf{x}}$ and the function $\gamma_{\mathbf{x}}$ can be calculated analytically (see Section C). This leads to an explicit expression for the constant $\Lambda$:

$$\Lambda = \frac{\sigma_w^3 \sigma_b^{\frac{1}{d}}}{(2\pi)^2} \Gamma(\tfrac{d+1}{2}) \Gamma(\tfrac{d}{2}+1)^{-(1+\frac{1}{d})} \left\langle \mu(\mathbf{x})^{-\frac{1}{d+1}} (r(\mathbf{x}))^{\frac{d-1}{d+1}} \right\rangle_\mu^{1+\frac{1}{d}}, \tag{24}$$

where $\langle u(\mathbf{x}) \rangle_\mu$ denotes the integral $\int \mu(\mathbf{x})u(\mathbf{x})d\mathbf{x}$. In the experiment, the value $\langle u(\mathbf{x}) \rangle_\mu$ can be computed by averaging $u$ over a $\mu$-distributed data set. In Figure 2 we compare theoretical and numerical NTK eigenvalue distributions for several dimensions $d$ and data set sizes $M$.

## 5.2 The loss function

We extend now the arguments of Section 4.2 to derive the loss asymptotic (1). We consider two classes of functions $g$ representing the initial error (4).

**Scenario 1: a discontinuous function $g$.** We assume that $g$ is supported on a bounded subset $\Omega \subset \mathbb{R}^d$ with a smooth boundary $\partial\Omega$ so that $g$ has a discontinuity on this boundary but is smooth inside $\Omega$. An obvious example of $g$ is the indicator function of $\Omega$. In this scenario we obtain

$$L(t) \sim \int_{\partial\Omega} |\Delta g(\mathbf{x})|^2 (\mu(\mathbf{x})\widetilde{\theta}_{\mathbf{x}}(\mathbf{n}))^{-\frac{1}{d+\alpha}} dS \cdot \frac{1}{2\pi}\Gamma(\tfrac{1}{d+\alpha}+1) \cdot (2t)^{-\frac{1}{d+\alpha}}. \tag{25}$$

Here, $\mathbf{x}$ is the point on the surface $\partial\Omega$, $\mathbf{n}$ is the unit normal to $\partial\Omega$, and $\Delta g(\mathbf{x})$ is the size of the jump of $g$ at $\mathbf{x}$, given by the limit of $g(\mathbf{y})$ as $\mathbf{y}$ approaches $\mathbf{x}$ from inside $\Omega$. The eigenvalue, coefficient, and loss exponents in this scenario are, respectively,

$$\kappa = \tfrac{1}{d}, \quad \nu = 1 + \tfrac{\alpha}{d}, \quad \xi = \tfrac{\kappa}{\nu} = \tfrac{1}{d+\alpha}. \tag{26}$$

**Scenario 2: $g$ generated by a Gaussian process.** Suppose that $g$ is a realization of a Gaussian process with a covariance matrix $\Sigma(\mathbf{x}, \mathbf{x}') = \langle g(\mathbf{x})g(\mathbf{x}') \rangle$ and that $\Sigma$ has a homogeneous singularity $\zeta_{\mathbf{x}}$ of degree $\beta$ at the diagonal (in the sense of Eqs. (16),(17)). In this scenario we find

$$L(t) \sim \int_{\mathbb{R}^d} \int_{|\mathbf{n}|=1} \widetilde{\zeta}_{\mathbf{x}}(\mathbf{n})(\mu(\mathbf{x})\widetilde{\theta}_{\mathbf{x}}(\mathbf{n}))^{-\frac{\beta}{d+\alpha}} d\mathbf{x}dS \cdot \frac{1}{2(2\pi)^d\beta}\Gamma(\tfrac{\beta}{d+\alpha}+1) \cdot (2t)^{-\frac{\beta}{d+\alpha}}. \tag{27}$$

Here, $\widetilde{\zeta}_{\mathbf{x}}$ is the Fourier transform of $\zeta_{\mathbf{x}}$ (defined similarly to the Fourier transform $\widetilde{\theta}_{\mathbf{x}}$). The eigenvalue, coefficient, and loss exponents in this scenario are, respectively,

$$\kappa = \tfrac{\beta}{d}, \quad \nu = 1 + \tfrac{\alpha}{d}, \quad \xi = \tfrac{\kappa}{\nu} = \tfrac{\beta}{d+\alpha}. \tag{28}$$

In our experiments we model GP by a large network in the NTK regime. The corresponding covariance is analyzed in Section 5.1 and has the singularity degree $\beta = 3$.

We provide the full derivations of Eqs. (25) and (27) in Section D of SM, and sketch now the main ideas. Our general strategy is to complement the localized eigenvalue analysis of Section 4.2 by the analysis of expansion coefficients. Note, however, that the simple approximation of $\widetilde{\mathcal{A}}$ by a direct sum of localized operators $\widetilde{\mathcal{A}}|_{L^2(\Omega_m)}$ (as performed in [5] and sketched in Section 4.2) is "too rough" for the study of expansion coefficients (due to a stronger effect of boundary conditions in each $\Omega_m$). Accordingly, we replace this approximation by the short-time Fourier transform of $g$:

$$F(\mathbf{y}, \mathbf{k}) = (2\pi)^{-d/2} \int_{\mathbb{R}^d} g(\mathbf{x}) \omega(\mathbf{x} - \mathbf{y}) e^{-i\mathbf{k}\cdot\mathbf{x}} d\mathbf{x},$$

where $\omega$ is a window function such that $\int \omega^2 = 1$. The coefficient $F(\mathbf{y}, \mathbf{k})$ describes the component of $g$ having the wave number $\mathbf{k}$ and localized at $\mathbf{y}$. Then at large $t$, using the stationary phase method,

$$g_t(\mathbf{x}) = e^{-t\widetilde{\mathcal{A}}} g(\mathbf{x}) \sim (2\pi)^{-\frac{d}{2}} \int_{\mathbb{R}^d} \int_{\mathbb{R}^d} F(\mathbf{y}, \mathbf{k}) e^{-t\mu_\mathbf{x}\widetilde{\theta}_\mathbf{x}(\mathbf{k})} \omega(\mathbf{x} - \mathbf{y}) e^{i\mathbf{k}\cdot\mathbf{x}} d\mathbf{y} d\mathbf{k}.$$

The leading contribution to this integral comes from large $\mathbf{k}$. For such $\mathbf{k}$, we can write the coefficients $F(\mathbf{y}, \mathbf{k})$ in an asymptotic form, primarily determined by the singularities of $g$. By integrating out $\mathbf{y}$ and $\mathbf{k}$, we then arrive at the desired Eqs. (25) and (27).

The above argument establishes the loss asymptotic (1) while bypassing the computation of the asymptotic (8) of the expansion coefficients aligned with the sorted "global" eigenvalues. This latter asymptotic (including the coefficient $K$) can be found by a similar computation, see Section D.4.

# 6 Extensions

The power law asymptotic of eigenvalues obtained in section 5.1 was based on the analysis of diagonal singularity of the NTK in the setting of ReLU activation, shallow depth 2, and the NTK regime. We argue now that our general approach is not restricted to this narrow setting. To show this, we separately consider several modifications of the network from Section 5.1 and an application to MNIST (see Figure 3). In this section we mostly limit ourselves to describing final results, with the derivations provided in SM, Section E.

## 6.1 Activations of different smoothness

Let's consider a shallow network in NTK regime with activation function $\phi_q(z) = (z)_+^q$, $q > 0$ ("a ReLU with the altered smoothness $q$"). Similarly to the ReLU case, one can show that NTK in current setting has a singularity on the diagonal for all values of $q$ except half-integers $q = \frac{1}{2}, \frac{3}{2}, \ldots$. The leading singular term is

$$\Theta_q(\mathbf{x}, \mathbf{x}') = \frac{\sigma_w^2}{2\pi} r^q r'^q a_q \varphi^{2q-1} \tag{29}$$

$$a_q = \frac{\Gamma^2(q)\Gamma(\frac{1}{2} - q)}{\sqrt{\pi} 2^q} \tag{30}$$

Here $r, r', \varphi$ are the same as in section 5.

The singularity of NTK with degree $\alpha = 2q - 1$ implies the eigenvalue power law asymptotic (6) with the exponent $\nu_q = 1 + \frac{2q-1}{d}$. Thus, the singularity degree in the NTK is determined by the singularity degree of the activation function. The coefficient $\Lambda_q$ can be explicitly computed as

$$\Lambda_q = \sigma_w^{\alpha+2} \sigma_b^{\frac{\alpha}{d}} q^2 (2\pi)^{d+q-2} \frac{\Gamma\left(\frac{d+\alpha}{2}\right)\Gamma^2(q)}{\left(\Gamma(\frac{d}{2}+1)\right)^{\frac{d+\alpha}{d}}} \left\langle \mu(\mathbf{x})^{-\frac{\alpha}{d+\alpha}} r(\mathbf{x})^{\frac{2d-\alpha d-\alpha}{d+\alpha}} \right\rangle_\mu^{\frac{d+\alpha}{d}} \tag{31}$$

In Figure 3a we compare the theoretical and numerical eigenvalue distributions for several values of $d$ and $q$.

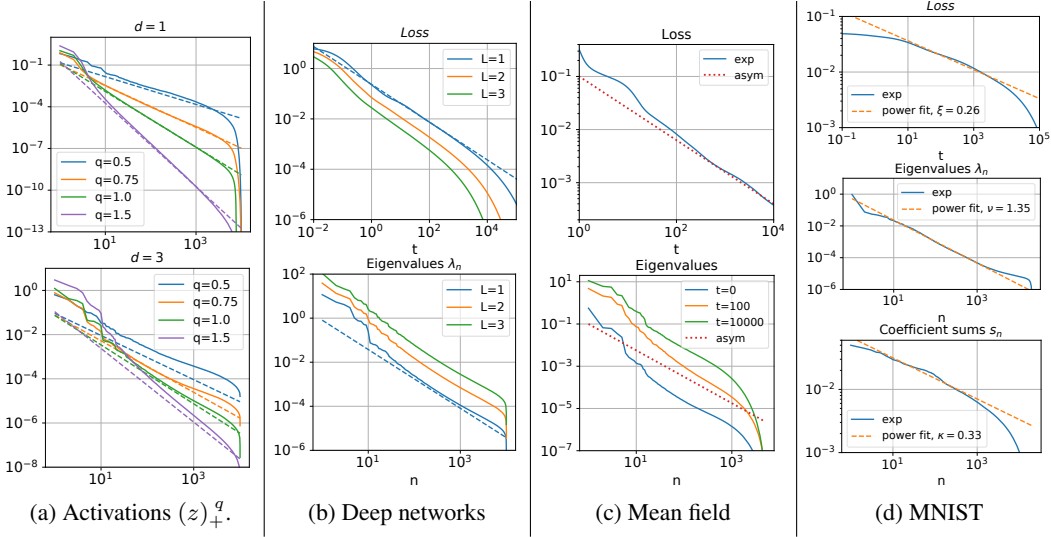

(a) Activations $(z)_+^q$.     (b) Deep networks     (c) Mean field     (d) MNIST

Figure 3: Extensions to other activations, deep networks, mean field regime, and MNIST.
**(a):** NTK eigenvalues for activation functions $\phi(z) = (z)_+^q$ with varying smoothness $q$. The theoretical distributions (dashed lines) have the exponents $\nu_q = 1 + \frac{2q-1}{d}$ (see Section 6.1).
**(b):** Loss evolution and NTK eigenvalue distribution for networks with varying number of hidden layers $L$ trained on 3-dimensional data and target generated by network GP ($\beta = 3$). In agreement with theory, the numerical results (solid lines) show the scaling exponents $\xi = \frac{3}{d+1}$ and $\nu = 1 + \frac{1}{d}$ for all depths $L$ (see Section 6.2). The theoretical predictions (dashed lines) are only shown for $L = 1$ (for other $L$ the theoretical lines would have the same slopes but different positions determined by the coefficients $C$ and $\Lambda$; due to computation complexity, we have found $C$ and $\Lambda$ only for $L = 1$).
**(c):** Loss dynamic and eigenvalue distribution at different moments of time for a network in the MF regime (Section 6.3). The network approximates GP on inputs with dimension $d = 4$. *Top:* comparison of the theoretical and experimental loss evolution. The theoretical exponent $\xi = \frac{\beta}{d+1} = \frac{3}{5}$. *Bottom:* the theoretical eigenvalue distribution and its experimental counterparts at different times. The theoretical exponent is $\nu = \frac{d+1}{d} = \frac{5}{4}$.
**(d):** Loss evolution, NTK eigenvalues $\lambda_n$ and coefficients $s_n$ for MNIST. All three curves are reasonably well approximated by power laws with exponents $\xi \approx 0.26$, $\nu \approx 1.35$, $\kappa \approx 0.33$ so that $\xi \approx \frac{\kappa}{\nu}$ in agreement with our theory. At the same time, our current theory does not predict specific values for the exponents $\nu$ and $\kappa$ because the intrinsic dimension of the MNIST manifold is very different from the large dimension of the ambient space. See SM, Section E for a detailed discussion.

## 6.2 Deep networks

We consider now a network of arbitrary depth $L > 2$ in the NTK regime and with the ReLU activation function. Similarly to the shallow ReLU case, the angles $\varphi_l$ (see Eq. (12)) are singular on the diagonal: $\varphi_l \propto |\mathbf{x} - \mathbf{x}'|$. Using relations (11) one can obtain recursive relations for $\varphi_l, r_l$ and finally for the singular part of NTK $\Theta_{\text{sing}}^{(l)}$

$$
\begin{aligned}
\Theta_{\text{diag}}^{(l+1)} &= r_{l+1}^2 + \frac{\sigma_w^2}{2}\Theta_{\text{diag}}^{(l)} \\
\Theta_{\text{sing}}^{(l+1)} &= -\frac{1}{2\pi}\Theta_{\text{diag}}^{(l)}\varphi_l + \frac{\sigma_w^2}{2}\Theta_{\text{sing}}^{(l)}
\end{aligned}
\tag{32}
$$

Here $\Theta_{\text{diag}}^{(l+1)}$ is the value of NTK on the diagonal. Since $\Theta_{\text{sing}}^{(1)} = 0$, we can see from (32) that $\Theta_{\text{sing}}^{(L)}$ is a weighted sum of $-\varphi_l$ for $l = 1, \ldots, L-1$. Thus, the singularity degree of the NTK is $\alpha = 1$ and the eigenvalue power law (6) holds with $\nu = 1 + \frac{1}{d}$. However, obtaining explicit formula for $\Lambda$ is harder for this case and we leave it for future work. In Figure 3b we compare the theoretical and numerical eigenvalue distributions for NTK's of deep networks.

### 6.3 MF regime

In this section we consider a shallow network in the MF regime and with the ReLU activation function, with the NTK given by (15). The neural tangent kernel (14) of the shallow NTK network can be obtained from (15) with the distribution $p_\infty(c, \mathbf{w}, b)$ taken as a product of Gaussians for each variable. Therefore, neural tangent kernels in MF regime represent a broader class of kernels, containing our basic example (14). It turns out that the diagonal singularity is present for all sufficiently smooth and quickly decaying distributions $p_\infty$ of this broader class:

$$\Theta_{\text{sing}}(\mathbf{x}, \mathbf{x}') = -\frac{\sqrt{1 + |\mathbf{x}|^2}}{2} \int\limits_{\mathbf{w} \cdot \mathbf{x} = -b} \left| \mathbf{w} \cdot \delta\mathbf{x} \right| dp_{\infty,2}(\mathbf{w}, b)$$

Here $\delta\mathbf{x} = \mathbf{x} - \mathbf{x}'$ and $p_{\infty,2}$ is the second moment of the distribution $p_\infty$ w.r.t. the variable $c$, i.e. $p_{\infty,2}(\mathbf{w}, b) = \int c^2 p_\infty(c, \mathbf{w}, b) dc$. Thus, the eigenvalue power law exponent is the same as in the NTK regime: $\nu = 1 + \frac{1}{d}$.

In Figure 3c we show the loss dynamic and the eigenvalue distribution at different moments of time for a network in the MF regime. Since the neural tangent kernel significantly changes during the training of the MF network, picking NTK's at different moments of training provides sufficiently distinct and general kernels of the form (15).

## 7 Discussion

The main results of the paper are theoretical derivations of power laws for the NTK spectrum and the loss asymptotic. Spectral properties of neural networks is an active area of current research [24, 34, 25, 28]. It has been observed (e.g., [32, 33, 14]) that, close to a global minimum, the Hessian of the loss of neural networks in practical tasks typically has eigenvalues accumulating near 0. This implies, in particular, that the Hessian is ill-conditioned, which has profound implications for network training: convergence of gradient descent in this case is much slower than for well-conditioned optimization problems. A power law spectral distribution is assumed in some recent works (e.g., for the study of generalization in [6]), but we are not aware of any previous general theoretical derivations of such a law. Our work essentially provides such a derivation (since at the global minimum the NTK spectrum coincides with the Hessian spectrum up to the eigenvalue 0). In a sense, we prove theoretically (under some reasonable assumptions) that a slow (power-law) convergence of the gradient descent is inevitable. Moreover, we precisely quantify this convergence.

Our work provides simple explicit formulas for the asymptotic eigenvalue distribution and loss evolution. This allows, in principle, to make some quantitative predictions about optimization not feasible with theoretical methods only providing upper bounds. For example, the loss exponent $\xi$ allows to directly estimate the number of GD steps needed to decrease the loss by one order of magnitude. In principle, the full loss asymptotic (25) can be used to analyze the effect of different data distributions $\mu(x)$ on convergence speed of GD.

The key idea of our asymptotic analysis is the extraction of the diagonal singularity from the NTK kernel. This is a new and promising way to relatively easily obtain information about the model spectrum: it only requires simple Taylor expansion and, as we show, is applicable in a wide range of scenarios, extending to deep networks, the mean field regime, and different activation functions. We expect this operation to be important for future studies of network training.

Our loss exponents $\xi = \frac{1}{\alpha+d}$ and $\xi = \frac{\beta}{\alpha+d}$ tend to 0 as the dimension $d$ increases. However, this does not contradict the practical trainability of networks for high-dimensional tasks such as image recognition, since real images occupy only tiny (i.e., effectively low-dimensional) subsets in the ambient high-dimensional space. While our current theory does not predict specific exponents in this setting, we do empirically observe power laws for eigenvalues, coefficients and the loss, and the fitted exponents satisfy our theoretical relation $\xi = \frac{\kappa}{\nu}$ (see Figure 3d). This suggests that our method can be extended to this setting, which will be the topic of a future work.

### Funding transparency statement

Research was supported by Russian Science Foundation, grant 21-11-00373.

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
