# Supplementary material

## A    Details of experiments

In this work we have two types of experiments. Both types operate with dataset consisting of $M$ data samples drawn from some distribution $\mu(\mathbf{x})$.

In our experiments we focus on distributions $\mu$ different from simple standard distributions such us spherical Gaussian or uniform in the cube. Highly symmetric distributions such as the spherical Gaussian make the whole problem analytically solvable (see, for example, [38]). In contrast, our theory do not rely on the symmetry, and we test it on distributions $\mu$ without any symmetry. In all our experiments, the distribution $\mu(\mathbf{x})$ is constructed as follows: we randomly choose $n_g$ points in the cube $[-1, 1]^d$ and consider $n_g$ symmetric Gaussian distributions with centers in these points and standard deviation $\sigma$; then $\mu(\mathbf{x})$ is defined as the average of these $n_g$ Gaussian distributions. Although each separate Gaussian distribution is symmetric, the average of $n_g > d$ randomly located Gaussian distributions almost surely removes all symmetries w.r.t. orthogonal transformations of $\mathbb{R}^d$. The typical values used in our experiments are $n_g = 8$ and $\sigma = 0.5$.

In the first type of experiments we analytically calculate the NTK using, for example, expression (14), and then numerically diagonalize the corresponding matrix and decompose a target function over its eigenvectors. Then the linear evolution $e^{-t\widetilde{A}}$ can be easily computed in obtained eigenbasis. The center and right parts of figure 1 as well as figures 2,3a,3b correspond to this type of experiments. Thus, it can be considered as an experiment with infinitely wide network, but a finite dataset. In all such experiments we take the largest $M$ possible, which is bounded by $O(M^3)$ time cost of numerical diagonalization and $O(M^2)$ memory cost of storing the NTK. The typical value used in experiments is $M = 10000$. To calculate the constant $\Lambda$ in the eigenvalue asymptotics (24) and (E.10) we draw another, rather big, set of points from the distribution $\mu(\mathbf{x})$ and use this dataset for Monte Carlo estimation of the $\langle \cdot \rangle_\mu$ averages in (24) and (E.10).

In the second type of experiments we initialize and train actual wide network (typical width $N = 3000$). The left part of figure 1 and figure 3c correspond to this type of experiments. To reach large values of time $t$ we choose the learning rate $\eta$ close to its critical value $\eta_c = \frac{2}{\lambda_0}$, above which the dynamic in the 0'th eigenspace start to diverge exponentially and the network leaves the regime of approximately constant NTK. The network in the MF regime adapts to learning rates higher than critical one at initialization, but adaptation resource is limited. Overall, experiments of this type test our theoretical predictions for roughly practical sizes of networks and datasets.

In the experiments we considered two types of target functions. The first is a draw from a Gaussian process, which we model by a very wide $N = 10^6$ shallow network with NTK parametrization. Thus, the covariance of GP is given by (13). To calculate the coefficient $C$ in the loss asymptotic (1) we use Eq. (27) with the sphere integral taken analytically as shown in (D.18). The second type of target is an indicator of a ball of some radius $r$. It corresponds to two-class classification task with first class located in $|\mathbf{x}| < r$ and the second class in $|\mathbf{x}| > r$. To calculate coefficient $C$ we sampled points uniformly on a sphere $|\mathbf{x}| = r$ and used them to calculate a Monte Carlo estimate of the integral in (25). In principle, one can choose classes with more sophisticated separation boundary, but numerical calculation of the integral in (25) will be more complicated. Note also that even if the target has a spherical symmetry, the whole problem does not, because we use a non-symmetric $\mu$.

## B Derivation of the loss asymptotic from the asymptotics of the eigenvalues $\lambda_n$ and the expansion coefficients $c_n$

In this section we prove the loss asymptotic (9). This result is established under assumption of power law asymptotics (6), (8) for the eigenvalues $\lambda_n$ and partial sums of coefficients $s_n = \sum_{k \geq n} |c_k|^2$, i.e.

$$\lambda_n \sim \Lambda n^{-\nu},$$
$$s_n \sim K n^{-\kappa}. \tag{B.1}$$

Here the asymptotic similarity sign $\sim$ denotes $a_n \sim b_n \iff a_n = b_n(1 + o(1))$.

**Theorem 1.** *Under the assumptions* (B.1) *on the asymptotic of eigenvalues ad coefficients, the loss* $L(t) = \frac{1}{2} \sum_n e^{-2\lambda_n t} |c_n|^2$ *has the asymptotic*

$$L(t) \sim \frac{K}{2} \Gamma \left( \frac{\kappa}{\nu} + 1 \right) (2\Lambda t)^{-\frac{\kappa}{\nu}} \tag{B.2}$$

*Proof.* The constant $\Lambda$ enters the loss only in combination with $t$. Thus, by rescaling time and noticing that the loss is proportional to $K$, it is sufficient to consider the case $K = 2$ and $2\Lambda = 1$. In other words, we have to prove

$$\sum_{n=0}^{\infty} e^{-tn^{-\nu}(1+u_n)}(s_n - s_{n+1}) \sim \Gamma \left( \frac{\kappa}{\nu} + 1 \right) t^{-\frac{\kappa}{\nu}}, \qquad \text{with} \quad s_n = n^{-\kappa}(1 + v_n) \tag{B.3}$$

Here $\lim_{n\to\infty} u_n = \lim_{n\to\infty} v_n = 0$ due to asymptotic (B.1).

The idea of the proof is that in the region of small $n$ the sum can be neglected due to exponential factor $e^{-t\#}$, while in the region of large $n$ the sum can be replaced by the integral, since the sum argument slowly depend on $n$ in this region. In fact both regions greatly overlap, and we can find a common point $n_t$ inside both regions. Such common point can be taken as $n_t = \lfloor t^\beta \rfloor$ with any $\beta$ from the interval $(\frac{1}{\nu+1}, \frac{1}{\nu})$ (the reason will be seen later). Let's denote

$$u_t \equiv \sup_{n \geq n_t} |u_n|, \qquad v_t \equiv \sup_{n \geq n_t} |v_n| \tag{B.4}$$

Since $u_n, v_n \to 0$ at $n \to \infty$ and $n_t \to \infty$ at $t \to \infty$, we have $u_t, v_t \to 0$ at $t \to \infty$. The strategy of the proof is first to bound the sum for $n \leq n_t$, then calculate the sum for $n > n_t$ with $u_n = v_n = 0$, and finally add corrections from $u_n$ and $v_n$.

The sum over $n \leq n_t$ is bounded as

$$\sum_{n=0}^{n_t} e^{-2t\lambda_n}(s_n - s_{n+1}) \leq e^{-2t\lambda_{n_t}} \sum_{n=0}^{n_t}(s_n - s_{n+1})$$

$$\leq \exp \left( -tn_t^{-\nu}(1 - u_t) \right) s_0 \leq \exp \left( -t^{1-\beta\nu}(1 - u_t) \right) s_0 \tag{B.5}$$

Since $\beta < \frac{1}{\nu}$ we have $1 - \beta\nu > 0$ and the sum goes to 0 exponentially fast as $t \to \infty$.

Now we calculate the sum over $n > n_t$ with $u_n = v_n = 0$. Due to convexity of $f(x) = x^{-\alpha}$ for all $\alpha > 0$ we have the bounds

$$\kappa(n+1)^{-\kappa-1} \leq s_n - s_{n+1} \leq \kappa n^{-\kappa-1} \tag{B.6}$$

Then we approximate the sum with the integral as

$$\sum_{n > n_t} e^{-tn^{-\nu}} \left( n^{-\kappa} - (n+1)^{-\kappa} \right) \stackrel{(1)}{=} \int_{n_t}^{\infty} e^{-tx^{-\nu}} \left( 1 + O(tx^{-\nu-1}) \right) \kappa x^{-\kappa-1} \left( 1 + O(x^{-1}) \right) dx$$

$$\stackrel{(2)}{=} \int_0^{tn_t^{-\nu}} e^{-z} \kappa \left( \frac{t}{z} \right)^{-\frac{\kappa+1}{\nu}} \frac{1}{t\nu} \left( \frac{t}{z} \right)^{1+\frac{1}{\nu}} dz \left( 1 + O(tn_t^{-\nu-1}) + O(n_t^{-1}) \right) \tag{B.7}$$

$$= \frac{\kappa}{\nu} t^{-\frac{\kappa}{\nu}} \left( \int_0^{\infty} z^{\frac{\kappa}{\nu}-1} e^{-z} dz + O\left( \exp\left( -tn_t^{-\nu} \right) [tn_t^{-\nu}]^{\frac{\kappa}{\nu}-1} \right) + O(tn_t^{-\nu-1}) + O(n_t^{-1}) \right)$$

$$\stackrel{(3)}{=} \Gamma \left( \frac{\kappa}{\nu} + 1 \right) t^{-\frac{\kappa}{\nu}} \left( 1 + O\left( \exp\left( -t^{1-\beta\nu} \right) [t^{(1-\beta\nu)}]^{\frac{\kappa}{\nu}-1} \right) + O\left( t^{1-\beta(\nu+1)} \right) + O\left( t^{-\beta} \right) \right)$$

Here in (1) we used (B.6) to estimate the difference $s_n - s_{n+1}$, and then first order Taylor expansion to estimate value of integrated function at non-integer points. In (2) we made a change of variables $z = tx^{-\nu}$ and estimated "big O" terms using minimum value $x_{\min} = n_t$. In (3) used the definition of Gamma function and recursive relation $z\Gamma(z) = \Gamma(z+1)$, and substituted $n_t$. $\beta \in (\frac{1}{\nu+1}, \frac{1}{\nu})$ implies that $1 - \beta\nu > 0$ and $1 - \beta(\nu + 1) < 0$, therefore all "big O" terms go to 0 as $t \to \infty$.

The last step is to include back $u_n$ and $v_n$ into sum over $n > n_t$. To include $u_n$ we use that $n^{-\nu}(1 - u_t) \le 2\lambda_n \le n^{-\nu}(1 + u_t)$. Then we make lower/upper bounds for the sum with $u_n$ by using the result of calculation (B.7) with substitution $t \to t(1 \pm u_t)$. Finally, we bound contribution from $v_n$ as

$$
\begin{aligned}
&\sum_{n>n_t} e^{-2t\lambda_n}\left(v_n n^{-\kappa} - v_{n+1}(n+1)^{-\kappa}\right) \\
&\stackrel{(1)}{=} \sum_{n>n_t}\left(e^{-2t\lambda_n} - e^{-2t\lambda_{n-1}}\right)v_n n^{-\kappa} + e^{-2t\lambda_{n_t}}v_{n_t+1}(n_t+1)^{-\kappa} \\
&\stackrel{(2)}{\le} v_t \sum_{n>n_t} e^{-2t\lambda_n}\left(n^{-\kappa} - (n+1)^{-\kappa}\right) + e^{-2t\lambda_{n_t}}(v_{n_t+1} - v_t)(n_t+1)^{-\kappa}
\end{aligned}
$$

(B.8)

Here in (1) we regrouped the terms in the sum, while in (2) we used $v_n \stackrel{n>n_t}{\le} v_t$ and regrouped summation terms back into original form. The second term in the last line is negative and the first term is $v_t$ times the result of (B.7). The final expression with all contributions is

$$
\begin{aligned}
\sum_{n=0}^{\infty} e^{-2t\lambda_n}(s_n - s_{n+1}) &= \Gamma\left(\frac{\kappa}{\nu} + 1\right)t^{-\frac{\kappa}{\nu}} \times \\
&\times \left(1 + O\left(\exp\left(-t^{1-\beta\nu}\right)\left[t^{(1-\beta\nu)}\right]^{\max(\frac{\kappa}{\nu}-1,0)}\right) + O\left(t^{1-\beta(\nu+1)}\right) + O(u_t) + O(v_t)\right)
\end{aligned}
$$

(B.9)

Here all "big O" terms vanish in the limit $t \to \infty$ and we obtain desired answer. $\qquad\square$

## C   Calculation of $\gamma_{\mathbf{x}}$

Following [5], given an even homogeneous function $\theta_{\mathbf{x}} : \mathbb{R}^d \to \mathbb{R}$ of degree $\alpha$, we define its Fourier transform $\widetilde{\theta}_{\mathbf{x}}$ using the Riesz summation formula:

$$\widetilde{\theta}_{\mathbf{x}}(\mathbf{k}) = \lim_{r \to \infty} \int_{\|\mathbf{x}\| < r} (1 - \tfrac{|\mathbf{x}|^2}{r^2})^c \theta(\mathbf{x}) e^{-i\mathbf{k}\cdot\mathbf{x}} d\mathbf{x}. \tag{C.1}$$

For sufficiently large $c$ the limit exists and is independent of $c$, and the resulting function $\widetilde{\theta}_{\mathbf{x}}$ is a homogeneous function of degree $-(d + \alpha)$.

According to formula (19) from the main text, to calculate the coefficient $\Lambda$ in the eigenvalue asymptotic one needs to find the volume $\gamma_{\mathbf{x}}$ of the region defined using $\widetilde{\theta}_{\mathbf{x}}(\mathbf{k})$:

$$\gamma_{\mathbf{x}} = (2\pi)^{-d} |\{\mathbf{k} \in \mathbb{R}^d : \widetilde{\theta}_{\mathbf{x}}(\mathbf{k}) > 1\}|. \tag{C.2}$$

In this section we calculate $\gamma_x$ in the case of singularities $\theta_{\mathbf{x}}(\mathbf{z}) = |\mathbf{z}|^\alpha$ and $\theta_{\mathbf{x}}(\mathbf{z}) = \varphi^\alpha(\mathbf{x}, \mathbf{x} + \mathbf{z})$ with $\varphi(\mathbf{x}, \mathbf{x}')$ defined as in the main text. The latter case is needed for obtaining coefficients $\Lambda$ in Eqs. (24) and (E.10).

**Case $\theta_{\mathbf{x}}(\mathbf{z}) = |\mathbf{z}|^\alpha$.**   We drop index $\mathbf{x}$ since there is no dependence on it. The homogeneity of $\theta(\mathbf{x})$ with degree $\alpha$ implies the homogeneity of $\widetilde{\theta}(\mathbf{k})$ with degree $-\alpha - d$. This can be seen from definition (C.1) by making integration variable change $\mathbf{x} \to c\mathbf{x}$, $c > 0$. Then, due to spherical symmetry of $\theta(\mathbf{z}) = |\mathbf{z}|^\alpha$, its Fourier transform has a form

$$\widetilde{\theta}(\mathbf{k}) = c_{d,\alpha} |\mathbf{k}|^{-d-\alpha} \tag{C.3}$$

To determine the coefficient $c_{d,\alpha}$ we take into account that both $\theta(\mathbf{z})$ and $\widetilde{\theta}(\mathbf{k})$ are generalized functions acting on some test functions $\chi$. We denote the Fourier transform by $F$ and action of generalized functions on test functions by $\langle \cdot, \cdot \rangle$ (not to be confused with averaging in Eqs. (24) and (E.10)). Then by taking a test function $\chi(\mathbf{k}) = e^{-|\mathbf{k}|^2/2}$ and its Fourier transform $\widetilde{\chi}(\mathbf{z}) = (2\pi)^{d/2} e^{-|\mathbf{z}|^2/2}$ we get

$$\langle F(\theta), \chi \rangle = c_{d,\alpha} \int |\mathbf{k}|^{-d-\alpha} e^{-\frac{|\mathbf{k}|^2}{2}} d\mathbf{k} = c_{d,\alpha} S_{d-1} \int_0^\infty k^{-1-\alpha} e^{-\frac{k^2}{2}} dk$$

$$= c_{d,\alpha} S_{d-1} 2^{-\frac{2+\alpha}{2}} \int_0^\infty e^{-x} x^{-\frac{2+\alpha}{2}} dx = c_{d,\alpha} S_{d-1} 2^{-\frac{2+\alpha}{2}} \Gamma\left(-\frac{\alpha}{2}\right) \tag{C.4}$$

$$\langle \theta, F(\chi) \rangle = (2\pi)^{\frac{d}{2}} \int |\mathbf{z}|^\alpha e^{-\frac{|\mathbf{z}|^2}{2}} d\mathbf{z} = (2\pi)^{\frac{d}{2}} S_{d-1} \int_0^\infty z^{d-1+\alpha} e^{-\frac{z^2}{2}} dz$$

$$= (2\pi)^{\frac{d}{2}} S_{d-1} 2^{\frac{d-2+\alpha}{2}} \int_0^\infty e^{-x} x^{\frac{d-2+\alpha}{2}} dx = (2\pi)^{\frac{d}{2}} S_{d-1} 2^{\frac{d-2+\alpha}{2}} \Gamma\left(\frac{d+\alpha}{2}\right) \tag{C.5}$$

Here in both calculations we first integrated over $d - 1$-dimensional sphere with the area $S_{d-1}$. Then we changed variables so that the integral can be expressed in terms of Gamma function. Since $\langle F(\theta), \chi \rangle = \langle \theta, F(\chi) \rangle$ we compare expressions (C.4) and (C.5) and find

$$c_{d,\alpha} = 2^{d+\alpha} \pi^{\frac{d}{2}} \frac{\Gamma\left(\frac{d+\alpha}{2}\right)}{\Gamma\left(-\frac{\alpha}{2}\right)} \tag{C.6}$$

To find $\gamma$ we notice that the volume in (C.2) is a ball with radius $c_{d,\alpha}^{1/(d+\alpha)}$. By using a volume of a unit ball in d-dimensional space $B_d = \pi^{d/2}/\Gamma\left(\frac{d}{2} + 1\right)$ we get

$$\gamma = \frac{1}{\Gamma\left(\frac{d}{2} + 1\right)} \left[ \frac{\Gamma\left(\frac{d+\alpha}{2}\right)}{\pi^{\frac{\alpha}{2}} \left| \Gamma\left(-\frac{\alpha}{2}\right) \right|} \right]^{\frac{d}{d+\alpha}} \equiv \gamma_{d,\alpha} \tag{C.7}$$

**Case $\theta_{\mathbf{x}}(\mathbf{z}) = \varphi^{\alpha}(\mathbf{x}, \mathbf{x} + \mathbf{z})$.** To see that angle $\varphi(\mathbf{x}, \mathbf{x}')$ has a singularity at $\mathbf{x} = \mathbf{x}'$ we write the scalar product of $\widetilde{\mathbf{x}}, \widetilde{\mathbf{x}}'$ as

$$\cos\varphi(\mathbf{x}, \mathbf{x}')\sqrt{r(\mathbf{x})r(\mathbf{x}')} = \sigma_w^2 \mathbf{x} \cdot \mathbf{x}' + \sigma_b^2 \tag{C.8}$$

with $r^2(\mathbf{x}) = \sigma_w^2|\mathbf{x}|^2 + \sigma_b^2$ as in the paper. Expanding this expression at $\varphi = 0$ and $\delta\mathbf{x} = \mathbf{x} - \mathbf{x}' = 0$ we get

$$\varphi(\mathbf{x}, \mathbf{x}') = \sqrt{1 - \frac{\sigma_w^2|\mathbf{x}|^2}{r^2(\mathbf{x})}\cos^2\psi} \, \frac{\sigma_w|\delta\mathbf{x}|}{r(\mathbf{x})} + O(|\delta\mathbf{x}|^2), \tag{C.9}$$

which is the expression (21) from the main text. In the asymptotic analysis we need to consider only the leading singular term, therefore the homogeneous singularity has the form

$$\theta_{\mathbf{x}}(\mathbf{z}) = a(\mathbf{x})\left(\sqrt{1 - b(\mathbf{x})\frac{z_1^2}{|\mathbf{z}|^2}}\,|\mathbf{z}|\right)^{\alpha} \tag{C.10}$$

Here $a(\mathbf{x}) = (\sigma_w/r(\mathbf{x}))^{\alpha}$ and $b(\mathbf{x}) = \sigma_w^2|\mathbf{x}|^2/r^2(\mathbf{x})$ are introduced for convenience and we also omit $\mathbf{x}$ for the rest of this section. In (C.10) we oriented basis in $\mathbf{z}$ space so that the first axis is parallel to vector $\mathbf{x}$. Now we calculate Fourier transform

$$\widetilde{\theta}(\mathbf{k}) = \int \theta(\mathbf{k})e^{-i\mathbf{k}\cdot\mathbf{z}}d\mathbf{z} = a\int\left[(1-b)z_1^2 + z_2^2 + \ldots + z_d^2\right]^{\frac{\alpha}{2}}e^{-i\mathbf{k}\cdot\mathbf{z}}d\mathbf{z}$$
$$\stackrel{(1)}{=} \frac{a}{\sqrt{1-b}}\int|\mathbf{z}'|^{\alpha}e^{-i\mathbf{k}'\cdot\mathbf{z}'}d\mathbf{z}' = \frac{a}{\sqrt{1-b}}c_{d,\alpha}|\mathbf{k}'|^{-d-\alpha} \tag{C.11}$$

Here in (1) we changed to variables $\mathbf{z}', \mathbf{k}'$ which are the same as $\mathbf{z}, \mathbf{k}$ except the first dimension: $z_1' = z_1\sqrt{1-b}$ and $k_1' = k_1/\sqrt{1-b}$. In the original $\mathbf{k}$ space the equation $c|\mathbf{k}'|^{-d-\alpha} = 1$ defines an ellipsoid obtained from the sphere $c|\mathbf{k}|^{-d-\alpha} = 1$ by squeezing the first axis by the factor $\sqrt{1-b}$. This gives us the formula for volume $\gamma$

$$\gamma = (2\pi)^{-d}B_d\left(\frac{c_{d,\alpha}a}{\sqrt{1-b}}\right)^{\frac{d}{d+\alpha}}\sqrt{1-b} = \gamma_{d,\alpha}a^{\frac{d}{d+\alpha}}\left(\sqrt{1-b}\right)^{\frac{\alpha}{d+\alpha}} \tag{C.12}$$

Restoring $\mathbf{x}$ dependence and using $\sqrt{1-b(\mathbf{x})} = \sigma_b/r(\mathbf{x})$ we get

$$\gamma_{\mathbf{x}} = \gamma_{d,\alpha}\sigma_w^{\frac{\alpha d}{d+\alpha}}\sigma_b^{\frac{\alpha}{d+\alpha}}r(\mathbf{x})^{-\frac{\alpha d+\alpha}{d+\alpha}} \tag{C.13}$$

**Case $\theta_{\mathbf{x}}(\mathbf{z}) = A(\mathbf{x})\varphi^{\alpha}(\mathbf{x}, \mathbf{x}+\mathbf{z})$.** This case includes the singularities of shallow network NTK (23) and covariance (22), and can be easily obtained from the previous one. Since Fourier transformation (C.1) and volume calculation (C.2) are performed locally, $A(\mathbf{x})$ is effectively constant in these calculations. Thus $\gamma_{\mathbf{x}}$ from (C.13) is simply multiplied by $|A(\mathbf{x})|^{\frac{d}{d+\alpha}}$. The result is

$$\gamma_{\mathbf{x}} = |A(\mathbf{x})|^{\frac{d}{d+\alpha}}\gamma_{d,\alpha}\sigma_w^{\frac{\alpha d}{d+\alpha}}\sigma_b^{\frac{\alpha}{d+\alpha}}r(\mathbf{x})^{-\frac{\alpha d+\alpha}{d+\alpha}} \tag{C.14}$$

Equation (24) can be obtained by using $A(\mathbf{x}) = -\frac{\sigma_w^2 r(\mathbf{x})^2}{2\pi}$ from (23) and then substituting resulting $\gamma_{\mathbf{x}}$ in (19).

# D   Derivation of the loss asymptotic for singular evolution operators $\widetilde{\mathcal{A}}$ and specific classes of target functions $g$

This section expands the content of Section 5.2 of the main text. Our goal is to derive the explicit leading terms in the asymptotic of the loss $L(t) = \frac{1}{2}\|g_t\|^2$ for the evolution $g_t = e^{-t\widetilde{\mathcal{A}}}g$ when $g$ belongs to one of the following two classes:

1. The function $g$ is supported on a bounded subset $\Omega$ with a smooth boundary $\partial\Omega$ so that $g$ has a discontinuity on this boundary but is smooth inside $\Omega$. An obvious example of $g$ is the indicator function of $\Omega$.

2. The function $g$ is a realization of a Gaussian process with a particular singular covariance.

Before providing these derivations, let us first recall our general setting. We consider the evolution $g_t = e^{-t\widetilde{\mathcal{A}}}g$ governed by the non-negative definite generator

$$\widetilde{\mathcal{A}}g(\mathbf{x}) = \int_{\mathbb{R}^d} \mu^{1/2}(\mathbf{x})\Theta(\mathbf{x},\mathbf{x}')\mu^{1/2}(\mathbf{x}')g(\mathbf{x}')d\mathbf{x}' \tag{D.1}$$

(see Section 3 of the main text). Here, $\mu$ is the measure corresponding to the data distribution, and $\Theta$ is the kernel associated with the neural network ansatz. This symmetric form of the evolution operator is valid in the representation in which the original functions are multiplied by $\mu^{1/2}$. Accordingly, the function $g$ appearing in the loss formula $L(t) = \frac{1}{2}\|e^{-t\widetilde{\mathcal{A}}}g\|^2$ is given by

$$g(\mathbf{x}) = \mu^{1/2}(\mathbf{x})(\widetilde{f}(\mathbf{w}(t=0),\mathbf{x}) - f(\mathbf{x})),$$

where $f$ is the function to be fitted by the network, and $\widetilde{f}(\mathbf{w}(t=0),\mathbf{x})$ is the initial network approximation.[1]

In scenario 1 above – a function $g$ supported on a domain $\Omega$ and discontinuous on the boundary $\partial\Omega$ – we will show that the large-$t$ asymptotic of the loss is given by

$$L(t) \sim \frac{1}{2\pi}\Gamma\left(\frac{1}{d+\alpha}+1\right)\int_{\partial\Omega}|\Delta g(\mathbf{x})|^2(\mu(\mathbf{x})\widetilde{\theta}_{\mathbf{x}}(\mathbf{n}))^{-\frac{1}{d+\alpha}}dS \cdot (2t)^{-\frac{1}{d+\alpha}}, \tag{D.2}$$

where integration is performed over the boundary, $\mathbf{x} \in \partial\Omega$ is the respective boundary point, $\mathbf{n}$ is the respective unit normal to the boundary, $\Delta g(\mathbf{x})$ is the value of discontinuity of $g$ at $\mathbf{x}$, and $\widetilde{\theta}_{\mathbf{x}}$ is the Fourier transform of the homogeneous singularity of the kernel $\Theta$ at $\mathbf{x} = \mathbf{x}'$.

In scenario 2 – a function $g$ generated by a Gaussian process with a homogeneous diagonal singularity of degree $\beta$ – we will show that the large-$t$ asymptotic of the loss is given by

$$L(t) \sim \frac{1}{2(2\pi)^d\beta}\Gamma\left(\frac{\beta}{d+\alpha}+1\right)\int_{\mathbb{R}^d}\int_{|\mathbf{n}|=1}\widetilde{\zeta}_{\mathbf{x}}(\mathbf{n})(\mu(\mathbf{x})\widetilde{\theta}_{\mathbf{x}}(\mathbf{n}))^{-\frac{\beta}{d+\alpha}}d\mathbf{x}dS \cdot (2t)^{-\frac{\beta}{d+\alpha}}, \tag{D.3}$$

where $\widetilde{\zeta}_{\mathbf{x}}$ is the Fourier transform of the diagonal singularity.

The general approach in obtaining these power law asymptotics is to expand $g$ over the approximate, spatially localized eigenvectors of the evolution operator $\widetilde{\mathcal{A}}$. One can consider two slightly different versions of this approach. In one version we first find the asymptotic (8) of the cumulative distribution of the expansion coefficients, and then find the asymptotic of the loss using formula (9) of the main text. In the other version, we bypass the computation of Eq. (8), and find the asymptotic of $L(t)$ directly. (In fact, this version also uses the asymptotic relation (9) of the main text, but applies it not to the full set of eigenvalues, but rather separately to the localized eigenvector expansion at each point $\mathbf{x}$ of the domain). We find the latter approach to be somewhat more direct and efficient.

Accordingly, our derivations will be structured as follows. In Section D.1 we discuss the general ideas of localization and high-frequency asymptotics. In Section D.2 we give a direct derivation for the loss asymptotic in the case of the first (discontinuous) class $g$. In Section D.3 we give a direct derivation for the loss asymptotic in the case of the second (Gaussian) class $g$. Then, in Section D.4 we sketch the derivation of the coefficient asymptotic (8) for both classes of $g$.

---

[1]We remark in passing that we can usually control the initial approximation $\widetilde{f}$, and in some cases we can ensure that the contribution of $\mu^{1/2}\widetilde{f}$ to $g$ is small compared to $\mu^{1/2}f$. In such cases, one can assume $g \approx -\mu^{1/2}f$.

## D.1 General considerations

Recall from Section 4.2 and 5.1 of the main text that the kernel $\Theta$ has a diagonal singularity:

$$\Theta(\mathbf{x}, \mathbf{x}') = \theta_\mathbf{x}(\mathbf{x} - \mathbf{x}') + \ldots, \tag{D.4}$$

where $\theta_\mathbf{x}(\cdot)$ is a (possibly $\mathbf{x}$-dependent) even ($\theta_\mathbf{x}(\mathbf{z}) = \theta_\mathbf{x}(-\mathbf{z})$) homogeneous function of degree $\alpha$:

$$\theta_\mathbf{x}(c\mathbf{z}) = |c|^\alpha \theta_\mathbf{x}(\mathbf{z}). \tag{D.5}$$

It will be convenient to also consider the function $\psi_\mathbf{x}$ obtained by rescaling $\theta_\mathbf{x}$ by the coefficient $\mu(\mathbf{x})$:

$$\psi_\mathbf{x} = \mu(\mathbf{x})\theta_\mathbf{x}.$$

We denote by $\widetilde{\theta}_\mathbf{x}$ and $\widetilde{\psi}_\mathbf{x}$ the versions of the Fourier transforms of $\theta_\mathbf{x}, \psi_\mathbf{x}$ defined as in Eq. (C.1). Note that these functions are homogeneous with degree $-(d + \alpha)$.

To analyze the evolution $g_t = e^{-t\widetilde{\mathcal{A}}}g$, we use the short-time Fourier transform (STFT) of $g$:

$$F(\mathbf{y}, \mathbf{k}) = (2\pi)^{-d/2} \int_{\mathbb{R}^d} g(\mathbf{x})\omega(\mathbf{x} - \mathbf{y})e^{-i\mathbf{k}\cdot\mathbf{x}}d\mathbf{x}, \tag{D.6}$$

$$g(\mathbf{x}) = (2\pi)^{-d/2} \int_{\mathbb{R}^d} \int_{\mathbb{R}^d} F(\mathbf{y}, \mathbf{k})\omega(\mathbf{x} - \mathbf{y})e^{i\mathbf{k}\cdot\mathbf{x}}d\mathbf{y}d\mathbf{k}. \tag{D.7}$$

Here $\omega$ is an even real smooth and compactly supported function such that $\int \omega^2 = 1$. Roughly speaking, the coefficient $F(\mathbf{y}, \mathbf{k})$ describes the component of $g$ having wave number $\mathbf{k}$ and localized at the point $\mathbf{y}$.

The stationary phase method (or its variant described in [5]) shows that if $f$ is a fixed function, then the leading term in the asymptotic of the action of the operator $\widetilde{\mathcal{A}}$ on the high-frequency function $f(\mathbf{x})e^{i\mathbf{k}\cdot\mathbf{x}}$ (with $\mathbf{k} \to \infty$) can be written in terms of the Fourier transform of the diagonal singularity of the kernel:

$$\int_{\mathbf{R}^d} \mu^{1/2}(\mathbf{x})\Theta(\mathbf{x}, \mathbf{x}')\mu^{1/2}(\mathbf{x}')f(\mathbf{x}')e^{i\mathbf{k}\cdot\mathbf{x}'}d\mathbf{x}' \sim \widetilde{\psi}_\mathbf{x}(\mathbf{k})f(\mathbf{x})e^{i\mathbf{k}\cdot\mathbf{x}}.$$

This shows that for large $\mathbf{k}$, we can think of the functions $f(\mathbf{x})e^{i\mathbf{k}\cdot\mathbf{x}}$ as of approximate eigenvectors of the operator $\widetilde{\mathcal{A}}$ and accordingly also of the evolution $e^{-t\widetilde{\mathcal{A}}}$. Then we can write

$$g_t(\mathbf{x}) = e^{-t\widetilde{\mathcal{A}}}g(\mathbf{x})$$

$$= (2\pi)^{-d/2} \int_{\mathbb{R}^d} \int_{\mathbb{R}^d} F(\mathbf{y}, \mathbf{k})e^{-t\widetilde{\mathcal{A}}}[\omega(\mathbf{x} - \mathbf{y})e^{i\mathbf{k}\cdot\mathbf{x}}]d\mathbf{y}d\mathbf{k}$$

$$\overset{|\mathbf{k}|\gg 1}{\sim} (2\pi)^{-d/2} \int_{\mathbb{R}^d} \int_{\mathbb{R}^d} F(\mathbf{y}, \mathbf{k})e^{-t\widetilde{\psi}_\mathbf{x}(\mathbf{k})}\omega(\mathbf{x} - \mathbf{y})e^{i\mathbf{k}\cdot\mathbf{x}}d\mathbf{y}d\mathbf{k}. \tag{D.8}$$

To justify the assumption $|\mathbf{k}| \gg 1$, observe that the function $e^{-t\widetilde{\psi}_\mathbf{x}(|\mathbf{k}|)}$ is close to 0 for $t|\mathbf{k}|^{-(d+\alpha)} \gg 1$ (i.e. for $|\mathbf{k}| \ll t^{\frac{1}{d+\alpha}}$), and is close to 1 for $t|\mathbf{k}|^{-(d+\alpha)} \ll 1$ (i.e. for $|\mathbf{k}| \gg t^{\frac{1}{d+\alpha}}$), so that at large $t$ the integral over $\mathbf{k}$ is indeed determined by the large-$|\mathbf{k}|$ asymptotic of $F(\mathbf{y}, \mathbf{k})$.

Next, we consider separately the two classes of functions $g$.

## D.2 Scenario 1: a discontinuous $g$

Suppose that $g$ is supported and smooth on the domain $\Omega$, and has a discontinuity $\Delta g$ at the boundary $\partial\Omega$. Consider the coefficient $F(\mathbf{y}, \mathbf{k})$ defined in Eq. (D.6). If $\mathbf{y}$ is such that the support of $\omega(\cdot - \mathbf{y})$ does not intersect the boundary $\partial\Omega$, then $g(\mathbf{x}')\omega(\mathbf{x}'-\mathbf{y})$ is a smooth function of $\mathbf{x}'$, and the coefficient $F(\mathbf{y}, \mathbf{k})$ will fall off faster than any power of $|\mathbf{k}|$ as $|\mathbf{k}| \to \infty$, so the contribution of such $\mathbf{y}$ to the expansion (D.8) will be negligible at large $t$. Assuming that the support of $\omega$ is small, it means that only $\mathbf{y}$ belonging to a narrow neighborhood of the boundary $\partial\Omega$ will contribute to (D.8). Accordingly, the function $g_t(\mathbf{x})$ will also fall off quickly away from the boundary.

Suppose now that $\mathbf{y}$ lies near the boundary and the support of $\omega(\cdot - \mathbf{y})$ intersects $\partial\Omega$. It is convenient to consider first the one-dimensional case $d = 1$.

**Case $d = 1$.** In this case the large-$k$ asymptotic of the coefficients $F(y, k)$ will be determined by the discontinuity of $g(x')\omega(x' - y)$ at the boundary point $x' = x_0 \in \partial\Omega$:

$$F(y, k) \sim (2\pi)^{-1/2} \frac{\Delta g(x_0)\omega(x_0 - y)}{ik} e^{-kx_0}.$$

Now we substitute this into Eq. (D.8):

$$g_t(x) = (2\pi)^{-1} \Delta g(x_0) \int_{\mathbb{R}} \int_{\mathbb{R}} \frac{e^{-t\widetilde{\psi}_{\mathbf{x}}(k)}}{ik} \omega(x_0 - y)\omega(x - y)e^{ik(x - x_0)} dy\, dk,$$

where $x_0$ is the point of $\partial\Omega$ near which the point $x$ is located[2]. Regarding the function $\omega(x - y)$, one observes by rescaling the variable $k$ that at large $t$, only a small (size-$t^{-\frac{1}{d+\alpha}}$) neighborhood of the boundary points will contribute to the integral, so we can write $\omega(x - y) \sim \omega(x_0 - y)$ and integrate out $y$ using the formula $\int_{\mathbb{R}} \omega^2 = 1$:

$$g_t(x) \sim (2\pi)^{-1} \Delta g(x_0) \int_{\mathbb{R}} \int_{\mathbb{R}} \frac{e^{-t\widetilde{\psi}_{\mathbf{x}}(k)}}{ik} \omega^2(x_0 - y)e^{ik(x - x_0)} dy\, dk$$

$$= (2\pi)^{-1} \Delta g(x_0) \int_{\mathbb{R}} \frac{e^{-t\widetilde{\psi}_{\mathbf{x}}(k)}}{ik} e^{ik(x - x_0)} dk. \tag{D.9}$$

We now want to estimate the full squared norm $\|g_t\|^2 = \int_{\mathbb{R}} |g_t(x)|^2 dx$. On the whole $\mathbb{R}$, the function $g_t$ is given by the sum of contributions from different points $x_0 \in \partial\Omega$:

$$g_t(x) \sim \sum_{x_0 \in \partial\Omega} (2\pi)^{-1} \Delta g(x_0) \int_{\mathbb{R}} \frac{e^{-t\widetilde{\psi}_{\mathbf{x}}(k)}}{ik} e^{ik(x - x_0)} dk. \tag{D.10}$$

Observe that the functions $u_{x_0}(k) = \frac{e^{-t\widetilde{\psi}_{\mathbf{x}}(k)}}{ik} e^{-ikx_0}$ with different $x_0 \in \partial\Omega$ become orthogonal in the limit $t \to \infty$ (since the Fourier transforms of these functions are localized at size-$t^{-\frac{1}{d+\alpha}}$ neighborhoods of the respective boundary points $x_0$). Then, by the unitarity of Fourier transform,

$$\|g_t\|^2 \sim \sum_{x_0 \in \partial\Omega} (2\pi)^{-1} |\Delta g(x_0)|^2 \int_{\mathbb{R}} \frac{e^{-2t\widetilde{\psi}_{\mathbf{x}}(k)}}{|k|^2} dk$$

$$= \sum_{x_0 \in \partial\Omega} |\Delta g(x_0)|^2 \frac{1}{\pi} \int_0^{\infty} \frac{\exp(-2\widetilde{\psi}_{\mathbf{x}}(1)tk^{-(d+\alpha)})}{|k|^2} dk.$$

The $t \to \infty$ asymptotic of the integral has been determined in Section B (see also formula (9) in the main text): setting $\nu = d + \alpha, \kappa = 1, \Lambda = \widetilde{\psi}_{\mathbf{x}}(1), K = 1$, we get

$$\|g_t\|^2 \sim \sum_{x_0 \in \partial\Omega} |\Delta g(x_0)|^2 \frac{1}{\pi} K\Gamma\left(\frac{\kappa}{\nu} + 1\right) (2\Lambda t)^{-\frac{\kappa}{\nu}}$$

$$= \sum_{x_0 \in \partial\Omega} |\Delta g(x_0)|^2 \frac{1}{\pi} \Gamma\left(\frac{1}{d + \alpha} + 1\right) (2\widetilde{\psi}_{\mathbf{x}}(1)t)^{-\frac{1}{d+\alpha}}.$$

**Case $d > 1$.** Assuming that the support of $\omega$ is small enough, we can approximate the boundary $\partial\Omega$ locally, in the support of $\omega(\cdot - \mathbf{y})$, by a linear hyperplane $\{\mathbf{x} : \mathbf{n} \cdot \mathbf{x} = x_0\}$, where $\mathbf{n}$ is the inward unit normal to $\partial\Omega$, so that the function $g(\cdot)\omega(\cdot - \mathbf{y})$ can be represented as a product of the Heaviside step function in the direction $\mathbf{n}$ and the smooth function $\omega(\cdot - \mathbf{y})$:

$$g(\mathbf{x}')\omega(\mathbf{x}' - \mathbf{y}) \sim \mathbf{1}_{\mathbf{n}\cdot\mathbf{x}'\geq x_0}(\mathbf{x}')\omega(\mathbf{x}' - \mathbf{y}).$$

The coefficient $F(\mathbf{y}, \mathbf{k})$ is the Fourier transform of this function w.r.t. $\mathbf{x}'$. The Fourier transform of $\mathbf{1}_{\mathbf{n}\cdot\mathbf{x}'\geq x_0}$ is a distribution concentrated on the line $l_{\mathbf{n}} = \{\mathbf{k} : \mathbf{k} = u\mathbf{n}, u \in \mathbb{R}\}$. The Fourier transform of $\mathbf{1}_{\mathbf{n}\cdot\mathbf{x}'\geq x_0}(\mathbf{x}')\omega(\mathbf{x}' - \mathbf{y})$ w.r.t. $\mathbf{x}'$ is the convolution of this line distribution with the Fourier transform of $\omega(\mathbf{x}' - \mathbf{y})$. Then for given $\mathbf{y}$, by the smoothness of $\omega$, the coefficients $F(\mathbf{y}, \mathbf{k})$

---

[2]For $d = 1$, we take $\Omega$ to be a finite union of intervals, so $\partial\Omega$ consists of finitely many points.

are concentrated in a neighborhood of the line $l_\mathbf{n}$ in the $\mathbf{k}$-space. Let $\mathbf{k}_\parallel$ denote the projection of vector $\mathbf{k}$ to this line. Since $\widetilde{\psi}_\mathbf{x}(\mathbf{k})$ is a homogeneous function, for large $t$ we can write $\widetilde{\psi}_\mathbf{x}(\mathbf{k}) \approx \widetilde{\psi}_\mathbf{x}(\mathbf{k}_\parallel)$ in the integral (D.8):

$$g_t(\mathbf{x}) \sim (2\pi)^{-d/2} \int_{\mathbb{R}^d} \int_{\mathbb{R}^d} F(\mathbf{y}, \mathbf{k}) e^{-t\widetilde{\psi}_\mathbf{x}(\mathbf{k}_\parallel)} \omega(\mathbf{x} - \mathbf{y}) e^{i\mathbf{k}\cdot\mathbf{x}} d\mathbf{y} d\mathbf{k}.$$

We can now decompose the wave number $\mathbf{k}$ and the vector $\mathbf{x}$ into components parallel and orthogonal to the normal $\mathbf{n}$ :

$$\mathbf{k} = \mathbf{k}_\parallel + \mathbf{k}_\perp = k_\parallel \mathbf{n} + \mathbf{k}_\perp, \quad \mathbf{x} = \mathbf{x}_\parallel + \mathbf{x}_\perp = x_\parallel \mathbf{n} + \mathbf{x}_\perp,$$

and perform integration over the component $\mathbf{k}_\perp$ in the above formula after substituting the expression for $F(\mathbf{y}, \mathbf{k})$:

$$g_t(\mathbf{x}) \sim (2\pi)^{-d} \int_{\mathbb{R}^d} \int_{\mathbb{R}^d} \int_{\mathbb{R}^d} g(\mathbf{x}') \mathbf{1}_{\mathbf{n}\cdot\mathbf{x}' \geq x_0}(\mathbf{x}') \omega(\mathbf{x}' - \mathbf{y}) e^{-t\widetilde{\psi}_\mathbf{x}(\mathbf{k}_\parallel)} \omega(\mathbf{x} - \mathbf{y}) e^{i\mathbf{k}\cdot(\mathbf{x}-\mathbf{x}')} d\mathbf{y} d\mathbf{k} d\mathbf{x}'$$

$$= (2\pi)^{-d+1} \int_{\mathbb{R}^d} \int_{\mathbb{R}} \int_{\mathbb{R}} g(\widetilde{\mathbf{x}}') \mathbf{1}_{x'_\parallel \geq x_0}(x'_\parallel) \omega(\widetilde{\mathbf{x}}' - \mathbf{y}) e^{-t\widetilde{\psi}_\mathbf{x}(\mathbf{k}_\parallel)} \omega(\mathbf{x} - \mathbf{y}) e^{ik_\parallel(x_\parallel - x'_\parallel)} d\mathbf{y} dk_\parallel dx'_\parallel,$$

where $\widetilde{\mathbf{x}}' = \mathbf{x}_\perp + x'_\parallel \mathbf{n}$. We can now proceed similarly to the previous case $d = 1$. Specifically, let $\mathbf{x}_0 = \mathbf{x}_\perp + x_0 \mathbf{n}$ be the projection of the point $\mathbf{x}'$ to the surface $\partial\Omega$. At large $t$ and $k_\parallel$, we can approximate $g(\widetilde{\mathbf{x}}') \approx g(\mathbf{x}_0), \widetilde{\mathbf{x}}' \approx \mathbf{x}_0, \mathbf{x} \approx \mathbf{x}_0$, and integrate out $\mathbf{y}$ using $\int \omega^2(\mathbf{x}_0 - \mathbf{y}) d\mathbf{y} = 1$ :

$$g_t(\mathbf{x}) \sim (2\pi)^{-d+1} \Delta g(\mathbf{x}_0) \int_\mathbb{R} \int_\mathbb{R} \mathbf{1}_{x'_\parallel \geq x_0}(x'_\parallel) e^{-t\widetilde{\psi}_\mathbf{x}(\mathbf{k}_\parallel)} e^{ik_\parallel(x_\parallel - x'_\parallel)} dk_\parallel dx'_\parallel$$

$$\sim (2\pi)^{-d} \Delta g(\mathbf{x}_0) \int_\mathbb{R} \frac{e^{-t\widetilde{\psi}_\mathbf{x}(\mathbf{k}_\parallel)}}{ik_\parallel} e^{ik_\parallel(x_\parallel - x_0)} dk_\parallel.$$

This expression is analogous to the expression (D.9), and similarly to the case $d = 1$, we can now use it to obtain the asymptotic of $\|g_t\|^2$. Recall that in the case $d = 1$, for each $x$ near the boundary we considered its projection $x_0$ to the boundary, and obtained the full integral $\|g_t\|^2$ by summing the contributions from different points $x_0$ (see Eq. (D.10) and subsequent formulas). In the present case $d > 1$, we replace this summation by integration over the boundary $\partial\Omega$. By repeating the same steps as before, we then get

$$\|g_t\|^2 \sim \frac{1}{\pi} \Gamma\left(\frac{1}{d+\alpha} + 1\right) \int_{\partial\Omega} |\Delta g(\mathbf{x})|^2 (2\widetilde{\psi}_\mathbf{x}(\mathbf{n}) t)^{-\frac{1}{d+\alpha}} dS$$

$$= \frac{1}{\pi} \Gamma\left(\frac{1}{d+\alpha} + 1\right) \int_{\partial\Omega} |\Delta g(\mathbf{x})|^2 (\mu(\mathbf{x})\widetilde{\theta}_\mathbf{x}(\mathbf{n}))^{-\frac{1}{d+\alpha}} dS \cdot (2t)^{-\frac{1}{d+\alpha}},$$

yielding the loss asymptotic (D.2).

## D.3    Scenario 2: $g$ generated by a Gaussian process

Suppose now that $g$ is generated by a Gaussian process with covariance $\Sigma(\mathbf{x}, \mathbf{x}') = \langle g(\mathbf{x}) g(\mathbf{x}') \rangle$, and that $\Sigma$ has a homogeneous singularity of degree $\beta$ on the diagonal $\mathbf{x} = \mathbf{x}'$ :

$$\Sigma(\mathbf{x}, \mathbf{x}') = \zeta_\mathbf{x}(\mathbf{x}' - \mathbf{x}) + \dots,$$

where the dots denote terms of a higher smoothness, and $\zeta_\mathbf{x}$ is an $\mathbf{x}$-dependent even homogeneous function of degree $\beta$ :

$$\zeta_\mathbf{x}(c\mathbf{z}) = |c|^\beta \zeta_\mathbf{x}(\mathbf{z}).$$

Similarly to the previously considered homogeneous functions, we denote by $\widetilde{\zeta}_{\mathbf{x}'}$ the Fourier transform of $\zeta_{\mathbf{x}'}$ defined using Eq. (26) of the main text.

To analyze the asymptotic of $\|g_t\|^2$, we use again the representation (D.8) in which we substitute the expansion for $F(\mathbf{y}, \mathbf{k})$:

$$g_t(\mathbf{x}) \sim (2\pi)^{-d} \int_{\mathbb{R}^d} \int_{\mathbb{R}^d} \int_{\mathbb{R}^d} g(\mathbf{x}') \omega(\mathbf{x}' - \mathbf{y}) e^{-t\widetilde{\psi}_\mathbf{x}(\mathbf{k})} \omega(\mathbf{x} - \mathbf{y}) e^{i\mathbf{k}\cdot(\mathbf{x}-\mathbf{x}')} d\mathbf{y} d\mathbf{k} d\mathbf{x}'. \quad \text{(D.11)}$$

Using as before the argument with rescaling, we see that the leading contribution to this integral comes at large $\mathbf{k}$ and small $\mathbf{x} - \mathbf{x}'$. In particular, we can write $\omega(\mathbf{x}' - \mathbf{y}) \approx \omega(\mathbf{x} - \mathbf{y})$ and integrate $\mathbf{y}$ out:

$$g_t(\mathbf{x}) \sim (2\pi)^{-d} \int_{\mathbb{R}^d} \int_{\mathbb{R}^d} g(\mathbf{x}') e^{-t\widetilde{\psi}_{\mathbf{x}'}(\mathbf{k})} e^{i\mathbf{k}\cdot(\mathbf{x}-\mathbf{x}')} d\mathbf{k} d\mathbf{x}'. \tag{D.12}$$

Also, for further convenience, we have replaced $\mathbf{x}$ by $\mathbf{x}'$ in $\widetilde{\psi}_{\mathbf{x}}(\mathbf{k})$.

We now approximate $\|g_t\|^2$ by its expectation over the target functions $g$ generated by the Gaussian process[3]:

$$\|g_t\|^2 \approx \langle \|g_t\|^2 \rangle \tag{D.13}$$

$$= \int_{\mathbb{R}^d} \langle g_t^2(\mathbf{x}) \rangle d\mathbf{x} \tag{D.14}$$

$$\sim (2\pi)^{-2d} \int_{\mathbb{R}^{5d}} \langle g(\mathbf{x}')g(\widetilde{\mathbf{x}}') \rangle e^{-t\widetilde{\psi}_{\mathbf{x}'}(\mathbf{k})} e^{i\mathbf{k}\cdot(\mathbf{x}-\mathbf{x}')} e^{-t\widetilde{\psi}_{\widetilde{\mathbf{x}}'}(\widetilde{\mathbf{k}})} e^{-i\widetilde{\mathbf{k}}\cdot(\mathbf{x}-\widetilde{\mathbf{x}}')} d\mathbf{k} d\mathbf{x}' d\widetilde{\mathbf{k}} d\widetilde{\mathbf{x}}' d\mathbf{x}. \tag{D.15}$$

We integrate out $\mathbf{x}$ using the identity $\int_{\mathbb{R}^d} e^{i(\mathbf{k}-\widetilde{\mathbf{k}})\cdot\mathbf{x}} d\mathbf{x} = (2\pi)^d \delta(\mathbf{k} - \widetilde{\mathbf{k}})$ :

$$\|g_t\|^2 \sim (2\pi)^{-d} \int_{\mathbb{R}^{3d}} \Sigma(\mathbf{x}', \widetilde{\mathbf{x}}') e^{-t\widetilde{\psi}_{\mathbf{x}'}(\mathbf{k}) - t\widetilde{\psi}_{\widetilde{\mathbf{x}}'}(\mathbf{k})} e^{i\mathbf{k}\cdot(\widetilde{\mathbf{x}}'-\mathbf{x}')} d\mathbf{k} d\mathbf{x}' d\widetilde{\mathbf{x}}'. \tag{D.16}$$

We isolate now the singularity and apply the stationary phase method, obtaining the high-frequency approximation

$$\int_{\mathbb{R}^d} \Sigma(\mathbf{x}', \widetilde{\mathbf{x}}') e^{-t\widetilde{\psi}_{\mathbf{x}'}(\mathbf{k}) - t\widetilde{\psi}_{\widetilde{\mathbf{x}}'}(\mathbf{k})} e^{i\mathbf{k}\cdot(\widetilde{\mathbf{x}}'-\mathbf{x}')} d\widetilde{\mathbf{x}}' \sim \widetilde{\zeta}_{\mathbf{x}'}(\mathbf{k}) e^{-2t\widetilde{\psi}_{\mathbf{x}'}(\mathbf{k})}, \quad |\mathbf{k}| \gg 1.$$

This leads to

$$\|g_t\|^2 \sim (2\pi)^{-d} \int_{\mathbb{R}^{2d}} \widetilde{\zeta}_{\mathbf{x}'}(\mathbf{k}) e^{-2t\widetilde{\psi}_{\mathbf{x}'}(\mathbf{k})} d\mathbf{k} d\mathbf{x}'. \tag{D.17}$$

To analyze the asymptotic of this integral at large $t$, we represent $\mathbf{k}$ as $|\mathbf{k}|\mathbf{n}$, where $\mathbf{n}$ is a unit vector. Then, using the large-$\mathbf{k}$ asymptotics $\widetilde{\zeta}_{\mathbf{x}'}(\mathbf{k}) = \widetilde{\zeta}_{\mathbf{x}'}(\mathbf{n})|\mathbf{k}|^{-(d+\beta)}$ and $\widetilde{\psi}_{\mathbf{x}'}(\mathbf{k}) = \widetilde{\psi}_{\mathbf{x}'}(\mathbf{n})|\mathbf{k}|^{-(d+\alpha)}$,

$$\|g_t\|^2 \sim (2\pi)^{-d} \int_{\mathbb{R}^d} \int_{|\mathbf{n}|=1} \int_0^\infty \widetilde{\zeta}_{\mathbf{x}'}(\mathbf{n}) r^{-(d+\beta)} e^{-2t\widetilde{\psi}_{\mathbf{x}'}(\mathbf{n})r^{-(d+\alpha)}} r^{d-1} d\mathbf{x}' dS dr$$

$$= (2\pi)^{-d} \int_{\mathbb{R}^d} \int_{|\mathbf{n}|=1} \widetilde{\zeta}_{\mathbf{x}'}(\mathbf{n}) \int_0^\infty r^{-(1+\beta)} e^{-2t\widetilde{\psi}_{\mathbf{x}'}(\mathbf{n})r^{-(d+\alpha)}} d\mathbf{x}' dS dr$$

$$\sim \frac{1}{(2\pi)^d \beta} \Gamma\left(\frac{\beta}{d+\alpha} + 1\right) \int_{\mathbb{R}^d} \int_{|\mathbf{n}|=1} \widetilde{\zeta}_{\mathbf{x}'}(\mathbf{n}) \widetilde{\psi}_{\mathbf{x}'}^{-\frac{\beta}{d+\alpha}}(\mathbf{n}) d\mathbf{x}' dS \cdot (2t)^{-\frac{\beta}{d+\alpha}}$$

$$= \frac{1}{(2\pi)^d \beta} \Gamma\left(\frac{\beta}{d+\alpha} + 1\right) \int_{\mathbb{R}^d} \int_{|\mathbf{n}|=1} \widetilde{\zeta}_{\mathbf{x}}(\mathbf{n}) (\mu(\mathbf{x})\widetilde{\theta}_{\mathbf{x}}(\mathbf{n}))^{-\frac{\beta}{d+\alpha}} d\mathbf{x} dS \cdot (2t)^{-\frac{\beta}{d+\alpha}},$$

which yields the loss asymptotic (D.3).

In the case when both GP and operator $\mathcal{A}$ originate from the same shallow ReLU network, Fourier transforms of diagonal singularities have similar angular dependence and integration over sphere can be performed analytically. We know from section C that if the kernel singularities are based on the angle $\varphi(\mathbf{x}, \mathbf{x}')$ between the input points $\mathbf{x}, \mathbf{x}'$, then $\widetilde{\zeta}_{\mathbf{x}}(\mathbf{z}) \propto \varphi^\beta(\mathbf{x}, \mathbf{x} + \mathbf{z})$ and $\widetilde{\theta}_{\mathbf{x}}(\mathbf{z}) \propto \varphi^\alpha(\mathbf{x}, \mathbf{x} + \mathbf{z})$. Then the respective Fourier transforms admit the forms $\widetilde{\zeta}(\mathbf{n}) = P(\mathbf{x})|\mathbf{n}'|^{-d-\beta}$ and

---

[3]In general (if the Gaussian process does not have a small correlation length), a sampled value of $\|g_t\|^2$ need not be close to the expectation $\langle \|g_t\|^2 \rangle$. However, one can show using the Wick-Isserlis formula that the variance $\langle (\|g_t\|^2 - \langle \|g_t\|^2 \rangle)^2 \rangle$ scales with $t$ as $t^{-(d+2\beta)/(d+\alpha)}$, i.e. becomes asymptotically negligible compared to $\langle \|g_t\|^2 \rangle^2$, which scales as $t^{-2\beta/(d+\alpha)}$. We plan to return to this point in a subsequent publication.

$\widetilde{\theta}(\mathbf{n}) = Q(\mathbf{x})|\mathbf{n}'|^{-d-\alpha}$, with $\mathbf{n}'$ the same as $\mathbf{n}$ except the first dimension: $n_1' = \frac{r(\mathbf{x})}{\sigma_b}n_1$. We write the sphere integral as

$$\int_{|\mathbf{n}|=1} dS|\mathbf{n}'|^{-d-\beta}\left(|\mathbf{n}'|^{-d-\alpha}\right)^{-\frac{\beta}{d+\alpha}} = \int_{|\mathbf{n}|=1} dS|\mathbf{n}'|^{-d}$$

$$= \int_{|\mathbf{n}|=1} dS\left(\frac{r^2}{\sigma_b^2}n_1^2 + (n_2^2 + \ldots + n_d^2)\right)^{-\frac{d}{2}}$$

$$\stackrel{(1)}{=} \int_0^\pi d\rho(\sin\rho)^{d-2}\int_{|\widetilde{\mathbf{n}}|=1} d\widetilde{S}\left(\frac{r^2}{\sigma_b^2}\cos^2(\rho) + \sin^2(\rho)\right)^{-\frac{d}{2}}$$

$$= S_{d-2}\int_0^\pi d\rho(\sin\rho)^{d-2}\left(\frac{r^2}{\sigma_b^2}\cos^2(\rho) + \sin^2(\rho)\right)^{-\frac{d}{2}}$$

$$= S_{d-2}\int_{-\infty}^{+\infty} d(\cot\rho)\left(\frac{r^2}{\sigma_b^2}\cot^2(\rho) + 1\right)^{-\frac{d}{2}}$$

$$= \frac{\sigma_b}{r}S_{d-2}\int_{-\infty}^{+\infty} dz(z^2+1)^{-\frac{d}{2}}$$

$$\stackrel{(2)}{=} \frac{\sigma_b}{r}S_{d-1} \tag{D.18}$$

Here in (1) we split integration over sphere $|\mathbf{n}| = 1$ over the first axis and remaining $d-2$ dimensional sphere $|\widetilde{\mathbf{n}}| = 1$: $n_1 = \cos\rho$ and $(n_2, \ldots, n_d) = \widetilde{\mathbf{n}}\sin\rho$. Finally in (2) the value of the integral over $z$ equals $S_{d-1}/S_{d-2}$, which can be inferred from the spherically symmetric case $r/\sigma_b = 1$.

### D.4 The coefficient distributions

The derivations given above bypass the explicit computation of the cumulative distribution function $s_n$ for the coefficients $c_n$ of the expansion of $g$ w.r.t. the eigenbasis of the operator $\widetilde{\mathcal{A}}$ (see Eqs. (7),(8)). These can be derived (at least heuristically) using essentially the same approach based on localized approximate eigendecomposition, but this time accompanied by the count of the total contribution of the coefficients corresponding to the given eigenvalue threshold from all the points of the domain.

It is convenient to introduce the partial sum $Q(\lambda)$ of the coefficients $|c_n|^2$ defined as in (7) but expressed in terms of the eigenvalue threshold $\lambda$:

$$Q(\lambda) = \sum_{n:\lambda_n<\lambda} |c_n|^2. \tag{D.19}$$

The large-$n$ asymptotic of $s_n$ corresponds to the small-$\lambda$ asymptotic of $Q(\lambda)$.

For Scenario 1 (discontinuous $g$), the resulting expression is

$$Q(\lambda) \sim \frac{1}{\pi}\int_{\partial\Omega} |\Delta g(\mathbf{x})|^2(\mu(\mathbf{x})\widetilde{\theta}_\mathbf{x}(\mathbf{n}))^{-\frac{1}{d+\alpha}} dS \cdot \lambda^{\frac{1}{d+\alpha}}. \tag{D.20}$$

For Scenario 2 (Gaussian $g$), the resulting expression is

$$Q(\lambda) \sim \frac{1}{(2\pi)^d\beta}\int_{\mathbb{R}^d}\int_{|\mathbf{n}|=1} \widetilde{\zeta}_\mathbf{x}(\mathbf{n})(\mu(\mathbf{x})\widetilde{\theta}_\mathbf{x}(\mathbf{n}))^{-\frac{\beta}{d+\alpha}} d\mathbf{x}dS \cdot \lambda^{\frac{\beta}{d+\alpha}}. \tag{D.21}$$

Since we have already found the loss asymptotics for both scenarios (Eqs. (D.2),(D.3)), we can establish the above expressions by showing that in either case $Q(\lambda) \sim a\lambda^b$ with some specific exponent $b$; the coefficient $a$ can then be deduced from the respective loss coefficient.

To find the exponent $b$, we consider again the STFT representation (D.6),(D.7). Suppose that the function $\omega$ lives on a small scale $\frac{1}{M}$:

$$\omega(\mathbf{x}) = M^{d/2}\omega_0(M\mathbf{x}),$$

and suppose that the domain is accordingly decomposed into $\propto M^d$ independent "$\mathbf{y}$–cells". We can think of the respective STFT coefficients $F(\mathbf{y}, \mathbf{k})$ as representing the actual coefficients in the eigenvector expansion. Consider now separately the two scenarios.

**Scenario 1: a discontinuous $g$.** The coefficients $F(\mathbf{y}, \mathbf{k})$ are negligible for cells not intersecting the boundary $\partial\Omega$. Suppose that the cell intersects $\partial\Omega$. Then the coefficients $F(\mathbf{y}, \mathbf{k})$ in this cell vanish outside the line $\mathbf{k} = u\mathbf{n}$ in the $\mathbf{k}$-space, where $\mathbf{n}$ is the unit normal to $\partial\Omega$. For $\mathbf{k} = u\mathbf{n}$, we have

$$|F(\mathbf{y}, \mathbf{k})| \propto M^{d/2} M^{1-d} \frac{|\Delta g(\mathbf{y})|}{|\mathbf{k}|} = M^{1-d/2} \frac{|\Delta g(\mathbf{y})|}{|\mathbf{k}|}.$$

We assume now that for suitable discrete wave numbers $\mathbf{k}$ the coefficients $F(\mathbf{y}, \mathbf{k})$ are associated to respective approximate eigenvectors of $\widetilde{\mathcal{A}}$ in the $\mathbf{y}$-cell, and estimate the respective contribution of the coefficients to the sum $S(\lambda)$. The discreteness results from the finite size of the support of $\omega$: the density of the eigenvalues scales with $M$ as $M^{-d}$. The respective discrete constants $u$ for the relation $\mathbf{k} = u\mathbf{n}$ scale as integer multiples of $M$, i.e. $u_l \sim lM$. Accordingly, the contribution of the coefficients $F(\mathbf{y}, \mathbf{k})$ in the $\mathbf{y}$-cell to $Q(\lambda)$ can be estimated as

$$\sum_{\mathbf{k}:\mathbf{k}=u_l\mathbf{n}, \lambda_{\mathbf{k}}<\lambda} |F(\mathbf{y}, \mathbf{k})|^2 \sim M^{-1} \int_{k_0}^{\infty} M^{2-d} \frac{|\Delta g|^2}{k^2} dk$$

$$\sim M^{1-d} \frac{|\Delta g|^2}{k_0}, \tag{D.22}$$

where the wave number $k_0 = |u\mathbf{n}|$ corresponds to the eigenvalue $\lambda$:

$$\widetilde{\theta}_{\mathbf{x}}(u\mathbf{n}) = \lambda.$$

We can find $k_0$ using the homogeneity of $\widetilde{\theta}_{\mathbf{x}}$: since $\widetilde{\theta}_{\mathbf{x}}(u\mathbf{n}) = |u|^{-d-\alpha}\widetilde{\theta}_{\mathbf{x}}(\mathbf{n})$, we have

$$k_0 = |u| = \left(\frac{\widetilde{\theta}_{\mathbf{x}}(\mathbf{n})}{\lambda}\right)^{1/(d+\alpha)}.$$

Substituting this into Eq. (D.22) and taking into account that there are $\propto M^{d-1}$ cells intersecting $\partial\Omega$, we find Eq. (D.20) up to a coefficient.

**Scenario 2: $g$ generated by a Gaussian process.** As before, finding the exponent $b$ can be reduced to estimating the asymptotic of $|F(\mathbf{y}, \mathbf{k})|^2$ at a fixed $\mathbf{y}$ and large $\mathbf{k}$. Computing the expectation, we get

$$\langle |F(\mathbf{y}, \mathbf{k})|^2 \rangle = (2\pi)^{-d} \int_{\mathbb{R}^d} \int_{\mathbb{R}^d} \omega(\mathbf{x} - \mathbf{y}) \omega(\mathbf{x}' - \mathbf{y})$$

$$\times \Sigma(\mathbf{x}, \mathbf{x}') e^{-i\mathbf{k}\cdot(\mathbf{x}-\mathbf{x}')} d\mathbf{x} d\mathbf{x}'$$

$$\propto |\mathbf{k}|^{-(d+\beta)}.$$

Since the wave vector $\mathbf{k}$ corresponds to an eigenvalue $\lambda \propto |\mathbf{k}|^{-(d+\alpha)}$ and since the density of the wave numbers $\mathbf{k} \in \mathbb{R}^d$ associated with localized eigenvectors of $\widetilde{\mathcal{A}}$ scales as $M^{-d}$, we can write

$$\sum_{\mathbf{k}:\lambda_{\mathbf{k}}<\lambda} |F(\mathbf{y}, \mathbf{k})|^2 \propto M^{-d} \int_{|\mathbf{k}|<\lambda^{-1/(d+\alpha)}} |\mathbf{k}|^{-(d+\beta)} d\mathbf{k}$$

$$\propto M^{-d} \lambda^{\beta/(d+\alpha)}.$$

Collecting the contributions to $Q(\lambda)$ from all $\sim M^d$ $\mathbf{y}$-cells, we thus get

$$Q(\lambda) \sim a\lambda^{\beta/(d+\alpha)},$$

as claimed.

By expressing $\lambda$ through $n$ with the help of the eigenvalue asymptotic (6) and Eq. (19), we can also cast the obtained formulas for $Q(\lambda)$ in the form $s_n \sim Kn^{-\kappa}$ as in Eq. (8).

For Scenario 1 (discontinuous $g$), the resulting coefficient and exponent are

$$\kappa = \frac{1}{d},$$

$$K = \left(\frac{1}{\pi} \int_{\partial\Omega} |\Delta g(\mathbf{x})|^2 (\mu(\mathbf{x})\widetilde{\theta}_{\mathbf{x}}(\mathbf{n}))^{-\frac{1}{d+\alpha}} dS\right) \left(\int \gamma_{\mathbf{x}} \mu^{\frac{d}{d+\alpha}}(\mathbf{x}) d\mathbf{x}\right)^{1/d}.$$

For Scenario 2 (Gaussian $g$), the resulting coefficient and exponent are

$$\kappa = \frac{1}{d},$$

$$K = \left(\frac{1}{(2\pi)^d\beta} \int_{\mathbb{R}^d} \int_{|\mathbf{n}|=1} \widetilde{\zeta}_{\mathbf{x}}(\mathbf{n})(\mu(\mathbf{x})\widetilde{\theta}_{\mathbf{x}}(\mathbf{n}))^{-\frac{\beta}{d+\alpha}} d\mathbf{x} dS\right) \left(\int \gamma_{\mathbf{x}} \mu^{\frac{d}{d+\alpha}}(\mathbf{x}) d\mathbf{x}\right)^{1/d}.$$

# E  Extensions

In this section we derive results of section 6 in the paper.

## E.1  Activations of different smoothness

We consider a shallow network in NTK regime with activation function $\phi_q(z) = (z)_+^q$, $q > 0$. Output covariance $\Sigma_q$ and NTK $\Theta_q$ for such network can be written as

$$\Sigma_q(\mathbf{x}, \mathbf{x}') = \sigma_w^2 \left\langle \left(z(\mathbf{x})\right)_+^q \left(z(\mathbf{x}')\right)_+^q \right\rangle \tag{E.1}$$

$$\Theta_q(\mathbf{x}, \mathbf{x}') = \Sigma_q(\mathbf{x}, \mathbf{x}') + \sigma_w^2 (\sigma_w^2 \mathbf{x} \cdot \mathbf{x}' + \sigma_b^2) q^2 \left\langle \left(z(\mathbf{x})\right)_+^{q-1} \left(z(\mathbf{x}')\right)_+^{q-1} \right\rangle \tag{E.2}$$

Here the average is taken w.r.t. pair of Gaussian random variables $z(\mathbf{x}), z(\mathbf{x}')$ with zero mean and covariance

$$\left\langle \left(z(\mathbf{x}), z(\mathbf{x}')\right)^T \left(z(\mathbf{x}), z(\mathbf{x}')\right) \right\rangle = \begin{pmatrix} r^2(\mathbf{x}) & r(\mathbf{x}) r(\mathbf{x}') \varphi(\mathbf{x}, \mathbf{x}') \\ r(\mathbf{x}) r(\mathbf{x}') \varphi(\mathbf{x}, \mathbf{x}') & r^2(\mathbf{x}') \end{pmatrix} \tag{E.3}$$

Such averages were calculated in [11] for integer $q$, but we take intermediate integral representation (eqs. (3), (16)) from this paper, which we will analyze for general $q$. As usual, we omit explicit $\mathbf{x}, \mathbf{x}'$ dependence for brevity.

$$\left\langle (z)_+^q (z')_+^q \right\rangle = \frac{1}{2\pi} r^q r'^q \Gamma(q+1) \left(\sin \varphi\right)^{2q+1} \int_0^{\frac{\pi}{2}} \frac{\left(\cos \psi\right)^q}{\left(1 - \cos \varphi \cos \psi\right)^{q+1}} d\psi \tag{E.4}$$

Let's denote the integral in (E.4) by $I_q(\varphi)$. We will transform it so it has the form of integral representation of the hypergeometric function $_2F_1$

$$
\begin{aligned}
I_q(\varphi) &= \int_0^{\frac{\pi}{2}} \frac{\left(\cos \psi\right)^q}{\left(1 - \cos \varphi \cos \psi\right)^{q+1}} d\psi \\
&= \int_0^1 \frac{y^q}{\sqrt{1 - y^2}(1 - \cos \varphi y)^{q+1}} dy, \quad y = \cos \psi \\
&= \frac{1}{(1 - \cos \varphi)^{q+1}} \int_0^1 t^q (1-t)^{-\frac{1}{2}} \left(1 + \frac{1 + \cos \varphi}{1 - \cos \varphi} t\right)^{-\frac{1}{2}} dt, \quad t = \frac{y(1 - \cos \varphi)}{1 - \cos \varphi y}
\end{aligned}
\tag{E.5}
$$

The hypergeometric function $_2F_1(a, b; c; z)$ has the following integral representation and asymptotic expansion at $z = -\infty$:

$$_2F_1(a, b; c; z) = \frac{\Gamma(c)}{\Gamma(b)\Gamma(c-b)} \int_0^1 t^{b-1}(1-t)^{c-b-1}(1-tz)^{-a} dt \tag{E.6}$$

$$_2F_1(a, b; c; -z) = z^{-b} \frac{\Gamma(a-b)\Gamma(c)}{\Gamma(a)\Gamma(-b+c)} \left(1 + \sum_{n \geq 1} g_n z^{-n}\right) + z^{-a} \sum_{n \geq 0} f_n z^{-n} \tag{E.7}$$

Here $g_n$ are $f_n$ are coefficients in the asymptotic expansion. Comparing our integral $I_q$ with integral representation of $_2F_1$ we see that it is indeed hypergeometric function with parameters $a = \frac{1}{2}, b = q+1, c = q + \frac{3}{2}$ and argument $z = -\frac{1 + \cos \varphi}{1 - \cos \varphi}$. Singularity at $\varphi = 0$ is located at hypergeometric function argument $z = -\infty$, therefore we need exactly asymptotic (E.7) to analyze singularity. Substituting our values of $_2F_1$ parameters we obtain the following asymptotic expansion

at $\varphi = 0$

$$(\sin\varphi)^{2q+1} I_q(\varphi) = \frac{\Gamma(q+1)\Gamma(-\frac{1}{2}-q)}{\sqrt{\pi}} \frac{(\sin\varphi)^{2q+1}}{(1+\cos\varphi)^{q+1}} \left[1 + \sum_{n\geq 1} g_n\left(\frac{1-\cos\varphi}{1+\cos\varphi}\right)^n\right]$$
$$+ \frac{(\sin\varphi)^{2q+1}}{(1-\cos\varphi)^{q+\frac{1}{2}}(1+\cos\varphi)^{\frac{1}{2}}} \sum_{n\geq 0} f'_n\left(\frac{1-\cos\varphi}{1+\cos\varphi}\right)^n \tag{E.8}$$

As it is written now, the asymptotic expansion above is not an expansion in powers $\varphi$, but it can be turned into one by replacing functions of $\varphi$ with their Taylor expansions. In particular, $\sin\varphi = \varphi + O(\varphi^3)$, $1 - \cos\varphi = \frac{1}{2}\varphi^2 + O(\varphi^4)$ and $1 + \cos\varphi = 2 + O(\varphi^2)$. In the asymptotic expansion (E.8) the second term starting from $\varphi^0$ is the leading one. However, it contains only even powers $\varphi^{2n}$, which are all regular. On the contrary, the first term starts with $\varphi^{2q+1}$ and it is singular for all $q$ except half-integers. Taking the leading singular term from (E.8) we obtain the leading singular term of NTK (E.2)

$$\Theta_{q,\text{sing}} = \frac{\sigma_w^2}{2\pi} r^{2q} q^2 \frac{\Gamma^2(q)\Gamma(\frac{1}{2}-q)}{\sqrt{\pi}2^q} \varphi^{2q-1} \tag{E.9}$$

Combining this with the $\gamma$ coefficient from (C.14) with $\alpha = 2q - 1$ and $A(\mathbf{x})$ deduced from (E.9) we get eigenvalue asymptotic coefficient $\Lambda_q$

$$\Lambda_q = \sigma_w^{\alpha+2} \sigma_b^{\frac{\alpha}{d}} q^2 (2\pi)^{d+q-2} \frac{\Gamma\left(\frac{d+\alpha}{2}\right)\Gamma^2\left(\frac{\alpha+1}{2}\right)}{\left(\Gamma(\frac{d}{2}+1)\right)^{\frac{d+\alpha}{d}}} \left\langle \mu(\mathbf{x})^{-\frac{\alpha}{d+\alpha}} r(\mathbf{x})^{\frac{2d-\alpha d-\alpha}{d+\alpha}} \right\rangle_\mu^{\frac{d+\alpha}{d}} \tag{E.10}$$

In the case of half-integer $q$ the coefficient in (E.9) diverges due to gamma function $\Gamma(\frac{1}{2}-q)$ having simple poles at positive half integer $q$. Quite interestingly, the same delta function is found in $\gamma_{d,\alpha}^{(d+\alpha)/d}$ and they cancel. Therefore, the final constant $\Lambda_q$ formally has a meaningful limit at half integer $q$. However, existence of a limit does not prove that at half-integer $q$ eigenvalues have an asymptotic with constant (E.10). The half integer case should be studied separately and we leave it for the future work.

## E.2 Deep networks

We consider deep network $L > 2$ in the NTK regime and with ReLU activation function. Covariances and NTK's of intermediate layers are calculated as

$$\begin{cases} \Sigma^{(l+1)}(\mathbf{x},\mathbf{x}') = \sigma_w^2 \langle \phi(z^l(\mathbf{x}))\phi(z^l(\mathbf{x}'))\rangle + \sigma_b^2 \\ \Theta^{(l+1)}(\mathbf{x},\mathbf{x}') = \Sigma^{(l+1)}(\mathbf{x},\mathbf{x}') + \sigma_w^2 \Theta^{(l)}(\mathbf{x},\mathbf{x}') \langle \dot\phi(z^l(\mathbf{x}))\dot\phi(z^l(\mathbf{x}'))\rangle \end{cases} \tag{E.11}$$

Here, as in the paper, $z^l(\mathbf{x})$ is a GP with covariance $\Sigma_l(\mathbf{x},\mathbf{x}')$. Parametrizing covariance as

$$\Sigma^{(l)}(\mathbf{x},\mathbf{x}') = \begin{pmatrix} r_l(\mathbf{x})^2 & r_l(\mathbf{x})r_l(\mathbf{x}')\cos\varphi_l(\mathbf{x},\mathbf{x}') \\ r_l(\mathbf{x})r_l(\mathbf{x})'\cos\varphi_l & r_l(\mathbf{x}')^2 \end{pmatrix} \tag{E.12}$$

From this point we again drop $\mathbf{x},\mathbf{x}'$ dependence. Using parametrization (E.12) we rewrite recursive relations (E.11)

$$\begin{cases} \Sigma^{(l+1)} = \frac{\sigma_w^2}{2\pi} r_l r'_l \left(\sin\varphi_l + \cos\varphi_l(\pi - \varphi_l)\right) + \sigma_b^2 \\ \Theta^{(l+1)} = \Sigma^{(l+1)} + \Theta^{(l)} \frac{\sigma_w^2}{2\pi}(\pi - \varphi_l) \end{cases} \tag{E.13}$$

We see that NTK's $\Theta^{(l)}$ can be fully expressed through $\varphi_l$ and $r_l$. From (E.13) the recursive relations for $\varphi_l, r_l$ are

$$\begin{cases} r_{l+1}^2 = \frac{\sigma_w^2}{2} r_l^2 + \sigma_b^2 \\ \cos\varphi_{l+1} = \frac{1}{r_{l+1}r'_{l+1}}\left[\frac{\sigma_w^2}{2\pi} r_l r'_l \left(\sin\varphi_l + \cos\varphi_l(\pi - \varphi_l)\right) + \sigma_b^2\right] \end{cases} \tag{E.14}$$

From these equations we see that $\varphi_{l+1} = 0$ only when $\varphi_l = 0$ and $r_l = r_l'$. This, in turn, happens only when $\mathbf{x} = \mathbf{x}'$. Using starting values $r_1^2(\mathbf{x}) = \sigma_w^2 |\mathbf{x}|^2 + \sigma_b^2$ and $\varphi(\mathbf{x}, \mathbf{x}')$ defined in (C.8), and the fact that $\arccos(z)$ is smooth everywhere except $z = -1, 1$, one can see that $r_l(\mathbf{x})$ is smooth everywhere and $\varphi(\mathbf{x}, \mathbf{x}')$ is smooth everywhere except the diagonal $\mathbf{x} = \mathbf{x}'$. Therefore, NTKs $\Theta^{(L)}$ are smooth away from diagonal and might have a singularity on it. From (E.13) and smoothness of $r_l(\mathbf{x})$ we see that the only source of singularity are $\varphi_l$.

To find singular expression for $\varphi_l$ let's assume that $\varphi_l = O(|\mathbf{x} - \mathbf{x}'|)$ and carefully expand second equation in (E.14) up to second order in $|\mathbf{x} - \mathbf{x}'|$. To do this we note that $|r_l - r_l'| = O(|\mathbf{x} - \mathbf{x}'|)$ and $\sin \varphi_l + \cos \varphi_l (\pi - \varphi_l) = \pi(1 - \varphi_l^2/2) + O(\varphi_l^3)$

$$\varphi_{l+1}^2 = \frac{\sigma_w^2 r_l^2}{2 r_{l+1}^2} \left( \varphi_l^2 + (r_l - r_l')^2 \frac{\sigma_b^2}{r_{l+1}^2} \right) + O(|\mathbf{x} - \mathbf{x}'|^3) \tag{E.15}$$

Thus we confirmed our assumption $\varphi_l = O(|\mathbf{x} - \mathbf{x}'|)$. In the leading order both $(r_l - r_l')^2$ and $\varphi_1^2$ are homogeneous functions of degree 2. Combining this with (E.15) we see that all $\varphi_l$ are homogeneous functions of degree 1 in the leading order of $\mathbf{x} - \mathbf{x}'$.

Now we find the leading singular part of $\Theta^{(l+1)}$. We will see that the leading singular part has degree 1, therefore we assume the following NTK expansion

$$\Theta^{(l)} = \Theta_{\text{diag}}^{(l)} - \sum_{m=1}^{l-1} a_m^{(l)} \varphi_m + O(|\mathbf{x} - \mathbf{x}|^2) \tag{E.16}$$

Here $\Theta_{\text{diag}}^{(l)}$ is the value of NTK at the diagonal, $a_m^{(l)}$ are constants and the sum represents leading singular part of the NTK $\Theta_{\text{sing}}^{(l)}$. The recursion relation (E.13) can be now written as

$$\Theta_{\text{diag}}^{(l+1)} + \Theta_{\text{sing}}^{(l+1)} + O(|\mathbf{x} - \mathbf{x}|^2) = r_{l+1}^2 + \left( \Theta_{\text{diag}}^{(l)} + \Theta_{\text{sing}}^{(l)} \right) \frac{\sigma_w^2}{2\pi} (\pi - \varphi_l) + O(|\mathbf{x} - \mathbf{x}|^2) \tag{E.17}$$

From this we extract recursive relations for diagonal and singular parts of NTK

$$\begin{aligned}
\Theta_{\text{diag}}^{(l+1)} &= r_{l+1}^2 + \frac{\sigma_w^2}{2} \Theta_{\text{diag}}^{(l)} \\
\Theta_{\text{sing}}^{(l+1)} &= -\frac{1}{2\pi} \Theta_{\text{diag}}^{(l)} \varphi_l + \frac{\sigma_w^2}{2} \Theta_{\text{sing}}^{(l)}
\end{aligned} \tag{E.18}$$

Constants $a_m^{(l)}$ can be explicitly extracted from this relations. Since $\Theta_{\text{diag}}^{(l)} > 0$ we can see that all $a_m^{(l)} > 0$. It means that the leading singular terms of order $O(|\mathbf{x} - \mathbf{x}|)$ will not cancel each over, thus confirming that the leading singularity in NTK has homogeneity degree 1.

### E.3   MF regime

The NTK of network in MF regime is given by

$$\Theta(\mathbf{x}, \mathbf{x}') = \int \nabla_{\tilde{\mathbf{w}}} \tilde{\phi}(\tilde{\mathbf{w}}, \mathbf{x}) p(\tilde{\mathbf{w}}) \nabla_{\tilde{\mathbf{w}}} \tilde{\phi}(\tilde{\mathbf{w}}, \mathbf{x}') d\tilde{\mathbf{w}} \tag{E.19}$$

Here $\tilde{\phi}(\tilde{\mathbf{w}}, \mathbf{x}) = c\phi(\mathbf{w} \cdot \mathbf{x} + b)$ is a computation of a single neuron in shallow network and $p(\tilde{\mathbf{w}}) = p(c, \mathbf{w}, b)$ is a distribution of a neuron parameters. In the case of ReLU activation $\phi(z) = (z)_+$ we rewrite it as

$$\begin{aligned}
\Theta(\mathbf{x}, \mathbf{x}') &= \int (\mathbf{w} \cdot \mathbf{x} + b)_+ (\mathbf{w} \cdot \mathbf{x}' + b)_+ dp_0(\mathbf{w}, b) \\
&\quad + (1 + \mathbf{x} \cdot \mathbf{x}') \int H(\mathbf{w} \cdot \mathbf{x} + b) H(\mathbf{w} \cdot \mathbf{x}' + b) dp_2(\mathbf{w}, b) \\
&= I_1(\mathbf{x}, \mathbf{x}') + (1 + \mathbf{x} \cdot \mathbf{x}') I_2(\mathbf{x}, \mathbf{x}')
\end{aligned} \tag{E.20}$$

Here the first integral $I_1$ corresponds to taking the gradient w.r.t. $c$ in (E.19) and the second integral $I_2$ corresponds to taking the gradients w.r.t. $\mathbf{w}$ and $b$; $p_0(\mathbf{w}, b)$ and $p_2(\mathbf{w}, b)$ are the 0'th and 2'nd moment of the distribution $p(c, \mathbf{w}, b)$ w.r.t. the variable $c$; $H(z)$ is the Heaviside step function.

Now suppose that $p_0$ and $p_2$ are sufficiently smooth and fall off quickly at infinity. To analyze the smoothness of the NTK (E.20) let's differentiate it using that $\frac{d}{dz}(z)_+ = H(z)$ and $\frac{d}{dz}H(z) = \delta(z)$. We start with the second integral $I_2$. The first derivative produces one delta function, and, together with the left Heaviside function, they are located on hyperplanes $\mathbf{w} \cdot \mathbf{x} + b = 0$ and $\mathbf{w} \cdot \mathbf{x}' + b = 0$ in the $(\mathbf{w}, b)$ space. First we consider a neighborhood of points $\mathbf{x} \neq \mathbf{x}'$ - the corresponding integral continuously depend on $\mathbf{x}, \mathbf{x}'$. If we differentiate the second time we will have two delta function, which restrict the integral to $d - 1$ dimensional subspace of $(\mathbf{w}, b)$ space, which is also continuously depend on $\mathbf{x}, \mathbf{x}'$. Further derivatives can be translated to differentiating $p_2(\mathbf{w}, b)$, with the result being continuous as long as $p_2$ is sufficiently smooth. Thus we established that the second integral is smooth at $\mathbf{x} \neq \mathbf{x}'$. Now let's turn to the diagonal $\mathbf{x} = \mathbf{x}'$, where two hyperplanes coincide, and consider the first derivative. Without loss of generality assume that the derivative is over $\mathbf{x}'$, then it is discontinuous, because infinitely small change of $\mathbf{x}$ will change which half of hyperplane $\mathbf{w} \cdot \mathbf{x}' + b = 0$ (corresponding to delta function) is located in the halfspace "allowed" by the Heaviside function: $\mathbf{w} \cdot \mathbf{x} + b > 0$. To summarize, $I_2(\mathbf{x}, \mathbf{x}')$ is smooth outside of the diagonal $\mathbf{x} = \mathbf{x}'$ and has a first order singularity on it. The first integral $I_1$ is treated similarly, except that one has to differentiate 3 times instead of 1.

Since the order of singularity is higher in $I_1$, the leading singular term comes from $I_2$. We focus now on deriving its behavior near the diagonal. Consider $\mathbf{x}' = \mathbf{x} + a\hat{\mathbf{n}}$ with small $a > 0$ and unit vector $\hat{\mathbf{n}}$. We write $I_2$ as

$$I_2(\mathbf{x}, \mathbf{x}') = \int\limits_{\mathbf{w} \cdot \mathbf{x} + b > 0} p_2(\mathbf{w}, b)d\mathbf{w}db \quad - \int\limits_{0 < \mathbf{w} \cdot \mathbf{x} + b < -a\mathbf{w} \cdot \hat{\mathbf{n}}} p_2(\mathbf{w}, b)d\mathbf{w}db \tag{E.21}$$

The first integral here is the values of $I_2$ on the diagonal, and the second integral is singular, because, e.g., it doesn't change sing with $\hat{\mathbf{n}} \to -\hat{\mathbf{n}}$ due to being non-negative. We calculate the second integral in (E.21) up to the first order in $|\mathbf{x} - \mathbf{x}'| = a$. The integration is taken in the thin region adjacent the the half of the hyperplane $\mathbf{w} \cdot \mathbf{x} + b = 0$ specified by $-\mathbf{w} \cdot \hat{\mathbf{n}} > 0$. The thickness of this region at point $\mathbf{w}$ can be calculated using geometric reasoning. The answer is $a(-\mathbf{w} \cdot \hat{\mathbf{n}})_+/\sqrt{|\mathbf{x}|^2 + 1}$. This gives us

$$I_2(\mathbf{x}, \mathbf{x}') - I_2(\mathbf{x}, \mathbf{x}) = -\frac{a}{\sqrt{|\mathbf{x}|^2 + 1}} \int\limits_{\mathbf{w} \cdot \mathbf{x} + b = 0} (-\mathbf{w} \cdot \hat{\mathbf{n}})_+ p_2(\mathbf{w}, b)d\mathbf{w}db + O(|\mathbf{x} - \mathbf{x}'|^2) \tag{E.22}$$

However, the expression (E.22) in principle contains both regular and singular parts. We can extract the singular part using the fact that it doesn't change sign under $\hat{\mathbf{n}} \to -\hat{\mathbf{n}}$, while the regular part does. Since the leading singular part of NTK comes from $I_2$, we can write it as

$$\Theta_{\text{sing}}(\mathbf{x}, \mathbf{x}') = \frac{\sqrt{|\mathbf{x}|^2 + 1}}{2} \int\limits_{\mathbf{w} \cdot \mathbf{x} + b = 0} |\mathbf{w} \cdot (\mathbf{x} - \mathbf{x}')| p_2(\mathbf{w}, b)d\mathbf{w}db. \tag{E.23}$$

We see that the singularity is of homogeneous type with degree 1, as for the network in NTK regime.

## E.4  MNIST and other high dimensional data

Almost all experimental tests presented in the main text involve relatively low-dimensional approximation problems (with $d$ ranging from 1 to 4). A natural question is whether the developed theory is applicable to high dimensional problems, such as image recognition and the likes, where the ambient dimension can be $10^3$ or higher. We should, however, point out a crucial difference between these problems and the setting of the present paper. Specifically, whereas in our setting we assume that the approximated function $f$ is defined on $\mathbb{R}^d$ or a $d$-dimensional subset thereof, in image recognition problems (and many other solvable high-dimensional problems) the function $f$ is only defined on a low-dimensional data manifold occupying a tiny portion in the ambient space. Accordingly, there are two completely different dimensions: the dimension of the ambient space (e.g., $28 \times 28$ for MNIST) and the much lower (approximate) intrinsic dimension of the data manifold. This data sparsity is quite a general property of solvable high-dimensional problems: without it, because of the curse of dimensionality, predictive models would need to be extremely complex. In contrast to the ambient dimension, the intrinsic dimension can only be roughly estimated by the metric properties of data manifold. In any case, we expect this dimension to be primarily associated with some natural

deformations of the considered objects (e.g., shifts, rotations, and other transformations of MNIST digits) and therefore to be not too large.

Since the data manifold can have a nontrivial curvature and other peculiarities not covered by the theoretical setting of the present paper, our theoretical predictions have only a limited applicability to this more general setting. A complete respective theory is beyond the scope of the present paper and will be a subject of a future study.

Nevertheless, in this paper we present experimental results on the eigenvalue, coefficient sum and loss asymptotics for MNIST (see Figure 3d). In this experiment, we used a shallow network in the NTK regime, with MSE loss and one-hot encoded classes as targets. The dataset size was 20000 points. We make several nontrivial observations about the experimental curves, suggesting that our method can indeed be generalized to a high-dimensional setting.

1. The eigenvalue and coefficient distributions $\lambda_n$ and $s_n$, as well as the loss function $L(t)$ are well approximated by the power laws (6),(8),(1) with exponents $\nu = 1.35, \kappa = 0.33, \xi = 0.26$.

2. The theoretically expected relation $\xi = \kappa/\nu$ is approximately satisfied for experimentally found values $\nu = 1.35, \kappa = 0.33, \xi = 0.26$. This is not very surprising since the derivation of the formula (9) is based on the assumptions (6) and (8), which are verified experimentally.

3. The third, and probably most interesting observation, concerns the effective dimension $d_{\mathrm{eff}}$ obtained by comparing the experimental values of the three exponents with the theoretical expressions (26), i.e. $\nu = 1 + \frac{1}{d_{\mathrm{eff}}}, \kappa = \frac{1}{d_{\mathrm{eff}}}, \xi = \frac{1}{d_{\mathrm{eff}}+1}$ (assuming they remain valid). A priori, the effective dimensions inferred from $\kappa$ and $\nu$ do not have to coincide because $\kappa$ and $\nu$ are not directly related. However, in our MNIST experiment, all three formulas give approximately the same value $d_{\mathrm{eff}} \approx 3$.

We leave a detailed study of these effects and their full theoretical explanation for a future work.

## F    Additional experiments

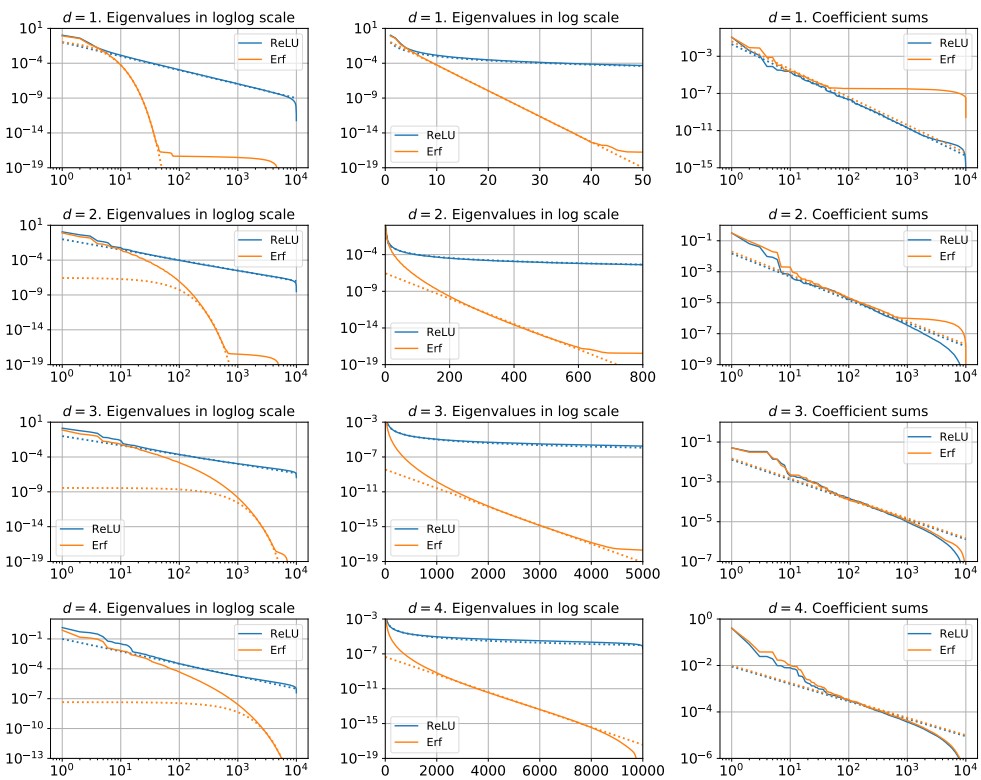

Figure 4: Distribution of eigenvalues $\lambda_n$ and coefficient partial sums $s_n$ for activation functions $\phi(z) = (z)_+$ and $\phi(z) = \mathrm{Erf}(z)$. Target functions in both cases are draws from Gaussian Process modeled by shallow network with NTK parametrization. Dotted lines show analytic expressions fitting the experimental distributions. Eigenvalues for the ReLU activation are fitted with the power law $\lambda_n = \Lambda n^{-1-\frac{1}{d}}$, eigenvalues for the Erf activation are fitted with the exponential law $\lambda_n \sim \Lambda e^{-an}$, and coefficients are fitted with the power law $s_n = K n^{-\frac{3}{d}}$.

In the paper we considered only discontinuous activation functions $\phi(z) = (z)_+^q$. The exponent in power law asymptotic in this case depends on activation smoothness $q$ as $\nu = 1 + \frac{2q-1}{d}$, which means that smoother activations produce NTKs with more quickly decreasing eigenvalues. The natural question is what would be the asymptotic of eigenvalues for smooth activation, although our theory does not apply to such activations. As an example of smooth activation we consider the error function $\phi(z) = \mathrm{Erf}(z)$. The NTK of shallow network with the error function activation is calculated in [21], and it is smooth everywhere.

In Figure 4 we can see eigenvalues $\lambda_n$ and coefficient partial sums $s_n$ for NTKs based on the ReLU and Erf activations. We see that the eigenvalues in the Erf case decrease much faster and can be approximately fitted by an exponential law $\lambda_n \sim \Lambda e^{-an}$. Quite interestingly, for both activations the coefficient distributions $s_n$ behave almost identically, which suggests that the eigenvectors are asymptotically represented by highly oscillating functions regardless of NTK type.

The second experiment is about data distribution. As we mentioned in Section A, the use of symmetric distributions $\mu(\mathbf{x})$ makes the evolution operator $\widetilde{\mathcal{A}}$ also symmetric. In Figure 5 we illustrate this point by plotting eigenvalue distribution for normal Gaussian distribution, uniform distribution on $[-1, 1]^d$ and average of randomly chosen Gaussians as described in Section A. We see that in the case of symmetric data distribution, the eigenvalue distribution has a staircase-like shape, especially for higher dimensions. This is explained by the high degeneracy of the eigenvalues of symmetric operators. However, for distribution $\mu(\mathbf{x})$ made of randomly chosen Gaussians the staircase-like

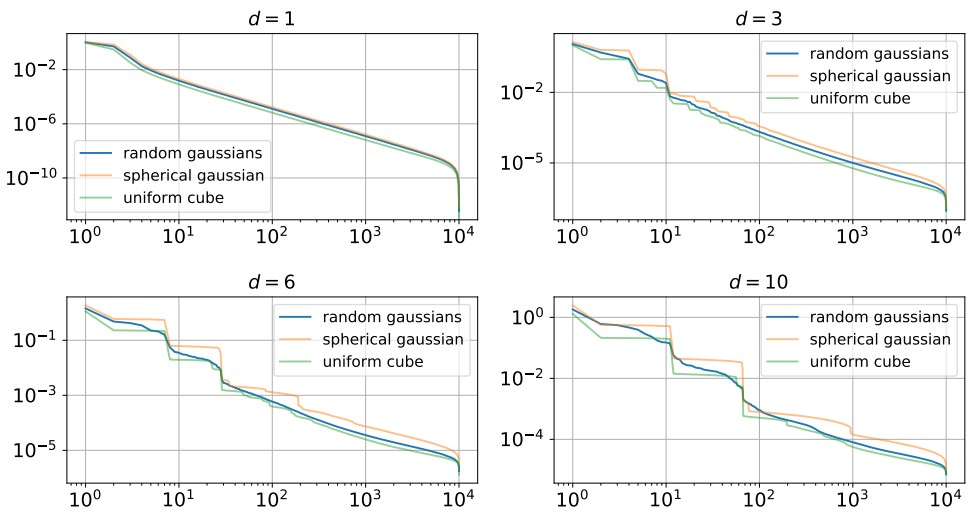

Figure 5: Eigenvalue distribution for different data distribution $\mu(\mathbf{x})$

shape is significantly smoothed, which indicates that such data distribution sufficiently eliminates all the symmetry-based features of corresponding linear operator.