# OpenReview forum: "Explicit loss asymptotics in the gradient descent training of neural networks"
_NeurIPS.cc/2021/Conference — NeurIPS 2021 Poster_

### Official Review · Reviewer_qRxk · 2021-07-12

**Rating:** 6
**Confidence:** 2

**Summary:**

The paper studies the scaling law in the training time of neural networks in the NTK regime.

**Limitations And Societal Impact:**

Sufficiently addressed.

**Main Review:**

The problem is novel and the scaling relations are very important both theoretically and practically. While the results in this work are interesting, I feel that the work could have spent some paragraphs discussing the practical uses of knowing the scaling law in training time.

Also, I am not too sure if the title is appropriate. The title reads: "Universal scaling laws in the gradient descent training of neural networks." However, the paper only studies the special form of scaling of the training loss with respect to the training time, but not all possible scaling laws. Therefore, I feel that the title needs to be changed to be more specific and avoid overclaiming.

I will consider raising the score if the authors discuss the practical side of the work in the rebuttal and propose a more appropriate title (or convince me why the present title is indeed precise and accurate).

Some other problems:
1. Figure 2 lacks axes labels.
2. line 158 "asisciate"




**Time Spent Reviewing:**

1.5 hour

---

> ### Author Response · Authors · 2021-08-10
> **Response to review**
>
> Thank you for your comments!
>
> Following your and another reviewer's suggestion, we decided to add a discussion section to our paper; please see the draft in the general response to reviewers above. This discussion would cover, in particular, some possible practical uses of our method.
>
> Regarding the practical uses of scaling law in training time, we think that such a simple form of training curve allows for a cheap estimation of the amount of computation (number of GD steps) needed to reach a certain loss value. Specifically, one can first run a few gradient descent steps ($\sim 10-100$ in our experiments) when the asymptotic is not yet applicable, and after that use the power law form of loss decay to forecast loss values at later training times.
>
> Regarding the title: actually, after submitting the paper, we also realized that the title does not quite reflect our specific contribution. We think a better title would be "Explicit loss asymptotics in the gradient descent training of neural networks".

---

> > ### Comment · Reviewer_ZGjL · 2021-08-18
> > **+1 on title**
> >
> > The new title is more appropriate in describing the contribution of the current paper.

---

> > ### Comment · Reviewer_qRxk · 2021-08-18
> > **update**
> >
> > Thanks for the additional discussion and the new title seems much better. I will raise the score from 5 to 6.

---

### Official Review · Reviewer_ZGjL · 2021-07-13

**Rating:** 6
**Confidence:** 4

**Summary:**

The paper studies the scaling law relation between $t$ (time) and the population loss in the NTK regime. The authors show that the scaling law is determined by two indices, (1) the index of the power law decay of the spectrum of the NTK (2) the index of the power decay of the residual of the loss on the eigenfunctions of the NTK.  The main technical contribution is the computation of the two indices.



**Limitations And Societal Impact:**

The work focuses on the NTK regime where the kernel is fixed, in which there is a nice and simple eigen-decomposition of the residual and does not cover the more realistic setting (mean field) where the kernel and the eigenvalues/eigenvectors are all evolving. Clearly, scaling law behavior under the latter setting is a very challenging open question in the field.

**Main Review:**

Scaling law behaviors of neural networks have attracted a lot of recent attention recently. The goal of this paper is study the scaling law under the NTK regime where the where the distribution of the learning task is given, in another words, the distribution of the input and the target are given and scaling law is mainly about the dependence of the population loss on time $t$. In this setting, population loss could be decompose into independent eigen-mode in the sense the loss is
$$
loss(t) = \sum_{n} e^{-2 \lambda_n t} g_n^2
$$ where $\lambda_n$ are the eigenvalues of the NTK and $g_n$ are the projections of the residual.  As such, the scaling law is determined by the index ($\nu$ in the paper) of the power law decay (assumed in the paper) of $\{\lambda_n\}$ and the index ($\beta$ in the paper) of the power law decay of the residual $\sum_{n\geq m} \|g_m|^2$. The paper then proceeds to compute these two indices under various of settings, which are the most technical part of the paper and which are also a very classical topic in mathematical analysis.

Overall, I think this is an interesting paper that draws a connection between scaling laws of neural network and oscillatory integrals although the analysis is limited to the NTK regime with popular loss.


**Time Spent Reviewing:**

5

---

> ### Author Response · Authors · 2021-08-10
> **Response to review**
>
> Thank you for the positive evaluation of our work!
>
> *The work focuses on the NTK regime where the kernel is fixed, in which there is a nice and simple eigen-decomposition of the residual and does not cover the more realistic setting (mean field) where the kernel and the eigenvalues/eigenvectors are all evolving*
>
> This is generally true, but please note a couple of things here.
>
> First, we do have some preliminary results for the mean field (Sections 6.3 and SM E.3); in particular we show that the eigenvalue exponent equals $1+\frac{1}{d}$ in this regime (like in the NTK regime).
>
> Second, while linearity indeed conveniently ensures existence of an eigendecomposition, we don't think that to actually find this decomposition is an easy task. We are not aware of any previous theoretical derivations of loss asymptotics with comparable generality and detail (even in a linearized setting). To the best of our knowledge, our general strategy of obtaining this asymptotic by extracting the singularity in the NTK and applying the stationary phase method is new.
>
> Third, recent empirical results [1] suggest that the difference between the NTK and classical regimes is not that drastic, and NTK models can even outperform finite networks on standard tasks such as CIFAR.
>
> Please also consider our general response to reviewers (with a draft of a new Discussion section).
>
> [1] Lee, J., Schoenholz, S., Pennington, J., Adlam, B., Xiao, L., Novak, R., & Sohl-Dickstein, J. (2020). Finite Versus Infinite Neural Networks: an Empirical Study. Advances in Neural Information Processing Systems, 33.

---

> > ### Comment · Reviewer_ZGjL · 2021-08-18
> > **update**
> >
> > There are a couple papers using the replica methods (non-rigorous) to compute the generalization analytically (scaling curves) of kernel methods; see e.g. [1]. The analysis is more technical since the setting is more difficult (test error vs number of sample.)
> >
> > I do agree that the current paper provides a simpler and more accessible explanation of the scaling low in the the population loss v.t. time setting, in which the distribution of the eigenvalues is all we need and which is a good starting point to understanding the scaling law in a more complex setting.
> >
> > >Third, recent empirical results [1] suggest that the difference between the NTK and classical regimes is not that drastic, and NTK models can even outperform finite networks on standard tasks such as CIFAR.
> >
> > Difference between NTK and classical regimes **CAN** be drastic, depending on the architectures. See CNN-VEC in Section B Main Table of SM.
> >
> > NTK outperforms finite width networks only in some non-interesting setting, e.g. fully-connected networks. For almost all useful architectures, e.g.  convolutional networks, finite width networks always outperform its NTK counter part. Moreover, intuition from small dataset regime (e.g. CIFAR10) does not extrapolate to larger data set regime (e.g. ImageNet), in which finite width networks are almost surely more useful.
> >
> > [1] Blake Bordelon, Abdulkadir Canatar, Cengiz Pehlevan.  Spectrum Dependent Learning Curves in Kernel Regression and Wide Neural Networks

---

> > > ### Author Response · Authors · 2021-08-19
> > > **comment to update**
> > >
> > > > There are a couple papers using the replica methods (non-rigorous) to compute the generalization analytically (scaling curves) of kernel methods; see e.g. [1]. The analysis is more technical since the setting is more difficult (test error vs number of sample.)
> > >
> > > Thanks for pointing out the generalization study [1]! We think that our work is complementary to this paper (and related papers of these authors).
> > >
> > > * We focus on the derivation of explicit spectral distributions, which is not the focus of the paper [1] (it rather considers the distribution as an assumption).
> > > * The only scenario in [1] in which the spectrum is derived is the (commonly considered) spherically symmetric case. This is a very special case, having features not found in "generic" problems --  e.g., a high degeneracy of eigenvalues; this degeneracy is not observed in realistic datasets such as MNIST.
> > > In contrast, we work in the setting where no symmetry is assumed. (We also have a brief discussion of the spherically symmetric case in SM, Section F.)
> > >
> > > In fact, one of the next things we plan to do is to combine our methods with the generalization analysis of the paper [1].

---

### Official Review · Reviewer_xC4H · 2021-07-14

**Rating:** 7
**Confidence:** 4

**Summary:**

The paper identifies a polynomial decay rate in the loss function during gradient descent under the condition that both the eigenvalues of the semigroup operator, and the coefficients in the eigenfunction expansion of an initial loss (g), exhibit power law decay. These assumptions are verified for the semigroup operator in the infinitely wide (NTK) regime, a mean field regime, and also, interestingly, under a homogeneity assumption in the transition kernel. Two scenarios for g are also considered. Throughout, the authors have taken care to be precise with coefficients and compute quantities exactly.

**Limitations And Societal Impact:**

The work presented is highly theoretical in nature. The limitations are predominantly to do with its assumptions, which are all clearly stated. Nevertheless, a concluding paragraph describing potential future improvements would be good to see.

**Main Review:**

I'm fond of what this paper manages to achieve. There is a great attention to detail from a mathematical point of view. The exposition is clear, assumptions well-specified, and a number of scenarios are covered. The recent push to consider spectral distributions in theoretical machine learning is an exciting one, and I think this paper is a prime example of what can be accomplished. An enjoyable read for sure.

However, much of the presentation seems too matter of fact for this venue, with little discussion surrounding consequences and interpretations. A concluding paragraph to interpret the findings more generally would be ideal. I recognize that this is hampered by space constraints. To cut down on space, there are a number of paragraphs that should probably belong in supplementary material instead. For this particular venue, lines 123-127 and 176-187 (for example) bog down the exposition. Otherwise, making more equations inline and cutting down on some unnecessary paragraph breaks could help with this.

As a side comment, there is a particular line of research establishing an empirical connection between tail exponents in power laws and generalization performance [1-3]. There has been a subsequent push to explain these connections theoretically. I believe the presented work could be an excellent contribution to this literature in the following sense: the belief to date is that decreasing the exponent nu in (6) should improve performance. This coincides well with the conclusion in (9). Furthermore, the relationship between nu, xi, alpha, and d, all seem to coincide well with empirical results. I recognize that the eigenspectrum of the operator A is not quite the same as the heavy-tailed distributions considered in the present literature, but I would expect there to be a relationship, due to the presence of the Jacobian in the NTK (2) and therefore in (3). Besides, much of the "heavy-tailed" literature is ad hoc about where the tail exponent comes from, since it is believed that many of the power laws encountered are related. As far as rate of convergence results are concerned, I also recall a similar strategy of using spectral distributions in [4] --- not sure if this is also worth mentioning.

With some of these changes, I believe this could be an outstanding paper --- at least a 7 or 8.

[1] Mahoney, M., & Martin, C. (2019). Traditional and heavy tailed self regularization in neural network models. In International Conference on Machine Learning (pp. 4284-4293). PMLR.

[2] Simsekli, U., Sagun, L., & Gurbuzbalaban, M. (2019). A tail-index analysis of stochastic gradient noise in deep neural networks. In International Conference on Machine Learning (pp. 5827-5837). PMLR.

[3] Martin, C. H., Peng, T. S., & Mahoney, M. W. (2021). Predicting trends in the quality of state-of-the-art neural networks without access to training or testing data. Nature Communications, 12(1), 1-13.

[4] Paquette, C., Lee, K., Pedregosa, F., & Paquette, E. (2021). SGD in the Large: Average-case Analysis, Asymptotics, and Stepsize Criticality. arXiv preprint arXiv:2102.04396.

**Time Spent Reviewing:**

6

---

> ### Author Response · Authors · 2021-08-10
> **Response to review**
>
> Thank you for the thorough reading of our work!
>
> Following your and another reviewer's suggestion, we decided to add a discussion section to our paper; please see the draft in the general response to reviewers above.
>
> In addition to the papers that you suggested (thank you!) we have found some other works which seem to be quite relevant to our research. The most straightforward connection concerns the Hessian of the loss, since at the global minimum its spectrum is identical, up to the eigenvalue 0, to the NTK spectrum that we study. Our results therefore directly complement those empirically analyzing the Hessian spectrum [5,6,7], or postulating its power law form (e.g., [8]). There might be other connections, for example we think that the exponent $\alpha$ describing the tail of the GD noise distribution in paper [2] may be connected to our exponents $\nu$ and $\kappa$.
>
> We have also noted that, as you point out, a lower eigenvalue exponents $\nu$ seems favorable for optimization. However, we are not yet ready to make strong claims about that, since we think a comprehensive analysis should also include effects of finite data set and network size, which were left outside of the present work. Since a kernel with a lower $\nu$ is more singular, it might, for example, require a larger network width for a valid approximation, thus offsetting the benefits of optimization speedup. We plan to look at this effect more closely in the future.
>
> [1-4] Provided by reviewer
>
> [5] Sagun, L., Bottou, L.,  LeCun, Y. (2016). Eigenvalues of the hessian in deep learning: Singularity and beyond.  arXiv:1611.07476.
>
> [6] Sagun, L., Evci, U., Guney, V. U., Dauphin, Y.,  Bottou, L. (2017). Empirical analysis of the hessian of over-parametrized neural networks.  arXiv:1706.04454.
>
> [7] Behrooz Ghorbani, Shankar Krishnan, Ying Xiao, An Investigation into Neural Net Optimization via Hessian Eigenvalue Density, ICML 2019
>
> [8] B. Bordelon, Abdulkadir Canatar, and C. Pehlevan.  Spectrum dependent learning curves in kernel regression and wide neural networks. ICML, 2020.

---

> > ### Comment · Reviewer_xC4H · 2021-08-23
> > **Thank you for your response**
> >
> > Thank you for responding to my comments and for the addition of a discussion section to the paper. I have read the discussion and feel it covers most of the right points. I understand the desire to leave out strong comments regarding the favourability of smaller $\nu$; indeed, it is possible that NTK approximation may be less effective in this regime. However, since the NTK approximation is justifiable in the very late stages of optimization, I would be surprised if this conclusion did not hold. Such a comprehensive analysis should focus on the small learning rate regime, as empirical evidence only seems to hold in this setting. Unfortunately, since the authors cannot reupload new versions of the paper, it is difficult for me to assess the changes in context, but I believe the paper should be accepted. I have increased my score to a 7.

---

### Official Review · Reviewer_ZAzG · 2021-07-16

**Rating:** 8
**Confidence:** 4

**Summary:**

This paper derives asymptotic scaling laws that characterize the rate of decay of the loss for neural networks trained by gradient descent in the NTK regime. Interestingly, the exponent of the power law describing this decay depends only on a few natural parameters (properties of the target function, dimensional of the data, smoothness of the nonlinearity).

**Limitations And Societal Impact:**

The most obvious limitation in this work is the NTK assumption. The authors are well aware of this fact and have some suggestive results in the mean-field regime near the end of the paper.



**Main Review:**

Many of the current guarantees for training are difficult to operationalize because they are weak upper bounds on the evolution of the loss. The authors here take a distinct approach that focuses instead on networks in the NTK regime which are formally infinite width. In essence, the approach can be outlined as follows: initially one assumes that the parameters are close to optimal (always the case in the infinite width limit) and linearize the dynamics. If the data density is smooth and compactly supported then the spectrum of the integral operator $\tilde{\mathcal{A}}$ that acts on the residual will have a well-behaved spectrum. In some cases, the spectrum of this operator has a power law form, leading to an evolution equation for the loss that decays like $t^{-\xi}$.

I found the exposition throughout, especially in Sec. 3, to be quite clear and precise. Appendix D gives a very complete accounting of the different function classes considered throughout.

The numerical results seem consistent with but not conclusive for the picture the authors advocate.

Sec. 6.3 did not provide enough context to argue for the power law.

**Time Spent Reviewing:**

2.5

---

> ### Author Response · Authors · 2021-08-10
> **Response to review**
>
> Thank you for the very positive evaluation of our work!
>
> *The numerical results seem consistent with but not conclusive for the picture the authors advocate*
>
> Indeed, there is some discrepancy between theory and experiment in our plots. Some aspects of this discrepancy are easy to explain. Consider, for example, the eigenvalue distribution plots in Figure 2. For large eigenvalues, the discrepancy is due to the asymptotic nature of the theory. For very small eigenvalues, the discrepancy is due to the finite size of the experimental data set: we see that the agreement improves as the size $M$ increases.
>
> In the case of the loss asymptotics as in Figure 1, we think that the agreement is actually rather good despite the moderate network width (3000) and data size (10K).
>
> *Sec. 6.3 did not provide enough context to argue for the power law*
>
> Unfortunately, due to  space constraints, we had to defer all the arguments about mean field to supplementary materials, section E.3. Due to the essential nonlinearity of the MF regime, currently we have only been able to theoretically obtain the power law exponent for the NTK eigenvalues (at any given moment of training), but not for the loss evolution. But we do empirically observe the power law decay of the loss (Figure 3c).

---

### Author Response · Authors · 2021-08-10
**General response to reviewers**

Following the suggestion of several reviewers, we plan to add an additional Discussion section at the end of the paper (at the cost of cutting some technical parts).

--------- Draft of the new Discussion section -------------

The main results of the paper are theoretical derivations of power laws for the NTK spectrum and the loss asymptotic. Spectral properties of neural networks is an active area of current research [1-4]. It has been observed (e.g., [5,6,7]) that, close to a global minimum, the Hessian of the loss of neural networks in practical tasks typically has eigenvalues accumulating near 0. This implies, in particular, that the Hessian is ill-conditioned, which has profound implications for network training: convergence of gradient descent in this case is much slower than for well-conditioned optimization problems. A power law spectral distribution is assumed in some recent works (e.g., for the study of generalization in [8]), but we are not aware of any previous general (without strong symmetry assumptions) theoretical derivations of such a law. Our work essentially provides such a derivation, since at the global minimum the NTK spectrum coincides with the Hessian spectrum up to the eigenvalue 0. In a sense, we prove theoretically that a slow, power-law convergence of the gradient descent is inevitable. Moreover, we precisely quantify this convergence.

Our work provides simple explicit formulas for the asymptotic eigenvalue distribution and loss evolution. This allows, in principle, to make some quantitative predictions about optimization not feasible with theoretical methods only providing upper bounds. For example, the loss exponent $\xi$ allows to directly estimate the number of GD steps needed to decrease the loss by one order of magnitude. In principle, the full loss asymptotic (25) can be used to analyze the effect of different data distributions $\mu(x)$ on convergence speed of GD.

The key idea of our asymptotic analysis is the extraction of the diagonal singularity from the NTK kernel. This is a new and promising way to relatively easily obtain information about the model spectrum: it is based on a usual Taylor expansion and, as we show, is applicable in a wide range of scenarios, extending to deep networks, the mean field regime, and different activation functions. We expect this operation to be useful for future studies of network training.

Our loss exponents $\xi=\frac{1}{\alpha+d}$ and $\xi=\frac{\beta}{\alpha+d}$ tend to 0 as the dimension $d$ increases. However, this does not contradict the practical trainability of networks for high-dimensional tasks such as image recognition, since real images occupy only tiny (i.e., effectively low-dimensional) subsets in the ambient high-dimensional space. While our current theory does not predict specific exponents in this setting, we do empirically observe power laws for eigenvalues, coefficients and the loss, and the fitted exponents satisfy our theoretical relation $\xi=\frac{\kappa}{\nu}$ (see Figure 3d). This suggests that our method can be extended to this setting, which will be the topic of a future work.

[1] Mahoney, M., & Martin, C. (2019). Traditional and heavy tailed self regularization in neural network models. In International Conference on Machine Learning (pp. 4284-4293). PMLR.

[2] Simsekli, U., Sagun, L., & Gurbuzbalaban, M. (2019). A tail-index analysis of stochastic gradient noise in deep neural networks. In International Conference on Machine Learning (pp. 5827-5837). PMLR.

[3] Martin, C. H., Peng, T. S., & Mahoney, M. W. (2021). Predicting trends in the quality of state-of-the-art neural networks without access to training or testing data. Nature Communications, 12(1), 1-13.

[4] Paquette, C., Lee, K., Pedregosa, F., & Paquette, E. (2021). SGD in the Large: Average-case Analysis, Asymptotics, and Stepsize Criticality. arXiv preprint arXiv:2102.04396.

[5] Sagun, L., Bottou, L.,  LeCun, Y. (2016). Eigenvalues of the hessian in deep learning: Singularity and beyond.  arXiv:1611.07476.

[6] Sagun, L., Evci, U., Guney, V. U., Dauphin, Y.,  Bottou, L. (2017). Empirical analysis of the hessian of over-parametrized neural networks.  arXiv:1706.04454.

[7] Behrooz Ghorbani, Shankar Krishnan, Ying Xiao, An Investigation into Neural Net Optimization via Hessian Eigenvalue Density, ICML 2019

[8] B. Bordelon, Abdulkadir Canatar, and C. Pehlevan.  Spectrum dependent learning curves in kernel regression and wide neural networks. ICML, 2020.

---

### Decision · Program_Chairs · 2021-09-27

**Decision:**

Accept (Poster)

**Comment:**

Four reviewers recommend this paper for acceptance. One reviewer indicated that s/he is not as concerned with the NTK assumption and that this work could initiate some interesting follow-up studies on the benefit of certain power law scalings, and raised her/his initial score. Another reviewer raised her/his score as the authors sufficiently addressed his/her questions and proposed convincing improvements. Another reviewer concludes that the primary limitation of this work is the NTK assumption, but that the that the clarity of the submission is high compared with other submissions, and that the reviewers agree that the results are sufficiently interesting and could initiate a number of different inquiries both theoretical and numerical. In summary, the general consensus is for acceptance. I agree with this view and hence I am recommending this submission for acceptance. The reviewers made several suggestions and several improvements were proposed in the authors’ responses. I request the authors take these carefully into consideration when preparing the final manuscript.